# Molecular architecture underlying fluid absorption by the developing inner ear

Keiji Honda[1†], Sung Huhn Kim[2‡], Michael C Kelly[3], Joseph C Burns[3§], Laura Constance[2], Xiangming Li[2#], Fei Zhou[2], Michael Hoa[4], Matthew W Kelley[3], Philine Wangemann[2*], Robert J Morell[5], Andrew J Griffith[1*]

[1]Molecular Biology and Genetics Section, National Institute on Deafness and Other Communication Disorders, National Institutes of Health, Bethesda, United States; [2]Anatomy and Physiology Department, Kansas State University, Manhattan, United States; [3]Developmental Neuroscience Section, National Institute on Deafness and Other Communication Disorders, National Institutes of Health, Bethesda, United States; [4]Auditory Development and Restoration Program, National Institute on Deafness and Other Communication Disorders, National Institutes of Health, Bethesda, United States; [5]Genomics and Computational Biology Core, National Institute on Deafness and Other Communication Disorders, National Institutes of Health, Bethesda, United States

**\*For correspondence:**
wange@vet.k-state.edu (PW);
griffita@nidcd.nih.gov (AJG)

**Present address:** [†]Otolaryngology Department, Tsuchiura Kyodo General Hospital, Tsuchiura, Japan; [‡]Department of Otorhinolaryngology, Head and Neck Surgery, Yonsei University College of Medicine, Seoul, Korea; [§]Decibel Therapeutics, Cambridge, United States; [#]Technique R and D-Drug Substance, GlaxoSmithKline Vaccine, US R and D center, Maryland, United States

**Competing interests:** The authors declare that no competing interests exist.

**Abstract** Mutations of *SLC26A4* are a common cause of hearing loss associated with enlargement of the endolymphatic sac (EES). *Slc26a4* expression in the developing mouse endolymphatic sac is required for acquisition of normal inner ear structure and function. Here, we show that the mouse endolymphatic sac absorbs fluid in an SLC26A4-dependent fashion. Fluid absorption was sensitive to ouabain and gadolinium but insensitive to benzamil, bafilomycin and S3226. Single-cell RNA-seq analysis of pre- and postnatal endolymphatic sacs demonstrates two types of differentiated cells. Early ribosome-rich cells (RRCs) have a transcriptomic signature suggesting expression and secretion of extracellular proteins, while mature RRCs express genes implicated in innate immunity. The transcriptomic signature of mitochondria-rich cells (MRCs) indicates that they mediate vectorial ion transport. We propose a molecular mechanism for resorption of NaCl by MRCs during development, and conclude that disruption of this mechanism is the root cause of hearing loss associated with EES.
DOI: https://doi.org/10.7554/eLife.26851.001

## Introduction

Enlargement of the vestibular aqueduct (EVA; OMIM 600791) is the most commonly detected osseous malformation of the inner ear in children with sensorineural hearing loss (*Griffith and Wangemann, 2011*). The vestibular aqueduct is a bony canal containing the endolymphatic duct and part of the endolymphatic sac. The width of this bony canal is thought to be established during development when the width of its mesenchymal precursor is determined by the epithelial endolymphatic duct. Therefore, enlargement of the endolymphatic sac and duct (EES) is the soft tissue correlate of EVA. The enlargement of the bony structures persists throughout life and is a radiologic marker of the underlying molecular pathophysiologic process that occurred during embryogenesis. The enlargement of the vestibular aqueduct is not considered to be the direct cause of hearing loss (*Griffith and Wangemann, 2011*).

The mammalian inner ear includes the cochlea, the vestibular labyrinth with saccule, utricle and three orthogonally arranged semicircular canals, and the endolymphatic duct and sac (*Figure 1A*).

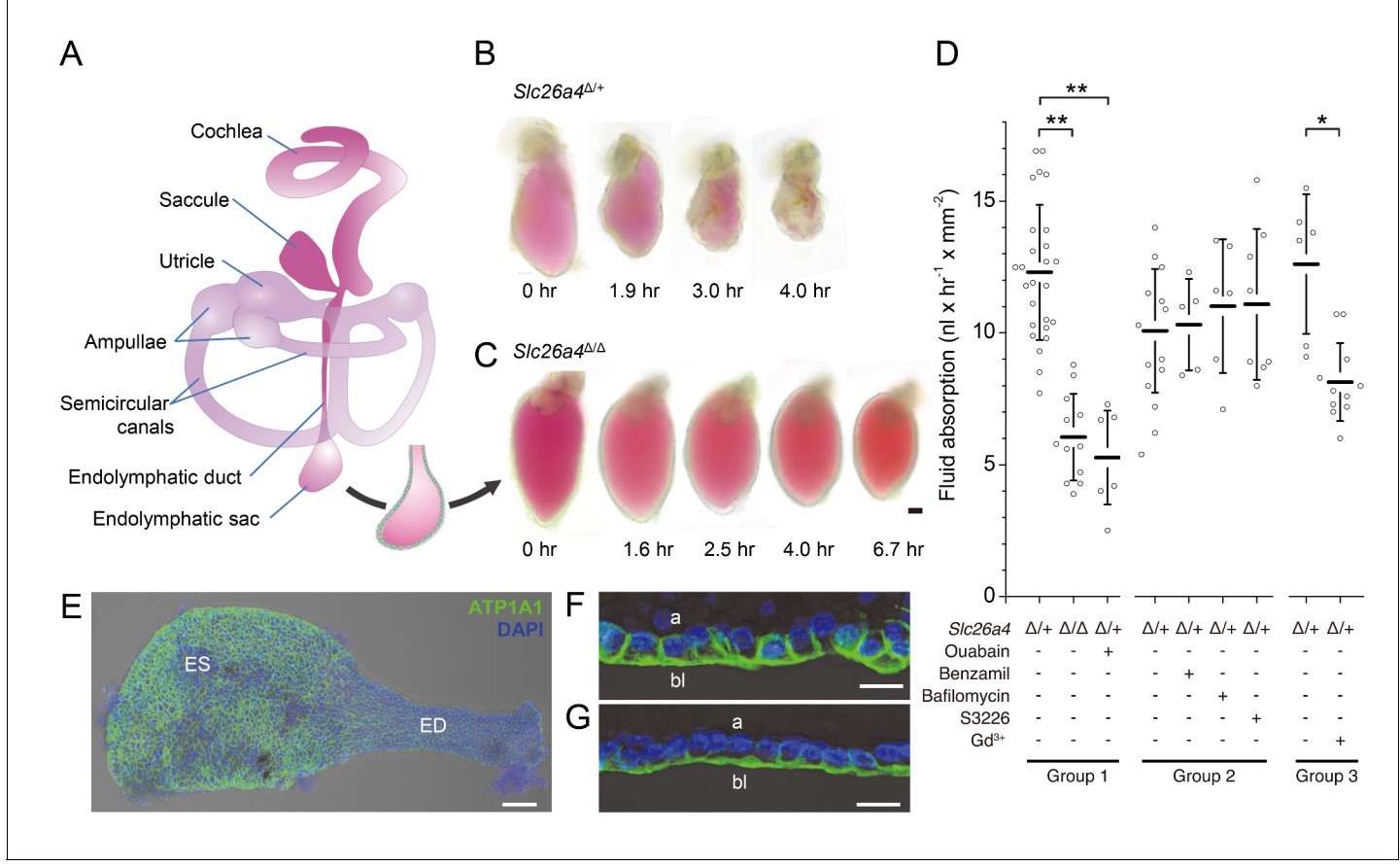

**Figure 1.** Fluid absorption in isolated endolymphatic sacs of $Slc26a4^{\Delta/+}$ and $Slc26a4^{\Delta/\Delta}$ mice. (A) Schematic diagram of the membranous labyrinth illustrating the location of the endolymphatic sac and other structures of the inner ear. (B–C) Endolymphatic sacs isolated from E14.5 $Slc26a4^{\Delta/+}$ and $Slc26a4^{\Delta/\Delta}$ mice were filled with the fluorescent dye SNARF-1 and repeatedly imaged by laser-scanning microscopy at the indicated time points. SNARF-1 has a strong absorption in the visible range causing the lumen to be colored red. Note that the sac from the $Slc26a4^{\Delta/+}$ mouse diminished in size faster than the sac from the $Slc26a4^{\Delta/\Delta}$ mouse. The luminal volume was recorded at each time point with 3D confocal microscopy to determine the rate of fluid absorption (*Figure 1—figure supplement 1A*). The bar represents 50 µm. (D) Summary of data obtained in three groups of experiments. Group 1: Rate of fluid absorption in E14.5 $Slc26a4^{\Delta/+}$ (n = 26), E14.5 $Slc26a4^{\Delta/\Delta}$ (n = 12) and E14.5 $Slc26a4^{\Delta/+}$endolymphatic sacs in the presence of ouabain (n = 7). Group 2: Rate of fluid absorption in E14.5 $Slc26a4^{\Delta/+}$endolymphatic sacs filled with control (n = 13), benzamil (n = 5), bafilomycin (n = 6) or S3226 solution (n = 8). Group 3: Rate of fluid absorption in E14.5 $Slc26a4^{\Delta/+}$endolymphatic sacs filled with control (n = 6) or gadolinium solution (n = 11). **p<0.001, *p<0.008. (E) Maximum intensity projection image of a whole-mounted E15.5 $Slc26a4^{\Delta/+}$ endolymphatic sac labeled by anti-ATP1A1 antibody (green) and stained with DAPI (blue). Note the higher levels of anti-ATP1A1 signal in endolymphatic sac (ES) compared to the endolymphatic duct (ED). Scale bar = 50 µm. (F,G) Optical cross-sections of the epithelium in the endolymphatic sac (F) and endolymphatic duct (G). Note that anti-ATP1A1 signal is located in basolateral (bl) but not apical (a) membranes. Bars represents 10 µm.

DOI: https://doi.org/10.7554/eLife.26851.002

The following source data and figure supplement are available for figure 1:

**Source data 1.** Rates of fluid absorption in endolymphatic sacs of E14.5 $Slc26a4^{\Delta/+}$ and $Slc26a4^{\Delta/\Delta}$ mice.
DOI: https://doi.org/10.7554/eLife.26851.004
**Figure supplement 1.** Geometric approximation of the luminal surface of isolated endolymphatic sacs.
DOI: https://doi.org/10.7554/eLife.26851.003

These structures comprise a continuous, fluid-filled epithelium called the membranous labyrinth. The cochlea, saccule, utricle and ampullae of the semicircular canals contain sensory hair cells. Cochlear hair cells detect sound. Vestibular hair cells in the saccule and utricle detect linear acceleration including gravity and vestibular hair cells in the ampullae of the semicircular canals detect rotational acceleration of the head. Although the endolymphatic sac lacks hair cells, it is an important structure that is essential for cochlear and vestibular function. The endolymphatic sac is an evolutionarily conserved structure present even in the primitive inner ear of hagfish (*Jørgensen et al., 1998*).

Moreover, the endolymphatic sac is the first structure to develop from the otocyst during early organogenesis (*Raft et al., 2014*; *Morsli et al., 1998*). These observations imply that the endolymphatic sac fulfills an important role in the development of the inner ear. Previous investigations suggest that the embryonic endolymphatic sac engages in NaCl resorption and that the ensuing fluid resorption is critical for morphogenesis of the inner ear and the progression of development leading to the acquisition of hearing and vestibular function (*Kim and Wangemann, 2010*; *Li et al., 2013a*).

Most investigations of the endolymphatic sac have focused on the function in the adult stage rather that during embryonic development. In adulthood, the endolymphatic sac appears to absorb endolymph and may thereby regulate endolymphatic pressure (*Salt, 2001*; *Inamoto et al., 2009*). The observations of proteins and related substances in the lumen of the endolymphatic sac had led to speculations that epithelial cells of the endolymphatic sac process these substances (*Rask-Andersen et al., 1999*; *Erwall et al., 2000*; *Guild, 1927*). Other investigations suggest that the endolymphatic sac contributes to immune responses (*Tomiyama and Harris, 1986*; *Wackym et al., 1987*).

The adult endolymphatic sac has been described as a simple cuboidal epithelium consisting of at least two cell types (*Dahlmann and von Düring, 1995*). Ribosome-rich cells (RRCs), also called dark cells, comprise a majority (~70%) of the epithelial cells. RRCs are characterized by an abundance of cytosolic ribosomes, rough endoplasmic reticulum, and cytoskeletal elements. Mitochondria-rich cells (MRCs), also called light cells, have plentiful mitochondria and numerous apical microvilli. The morphology of MRCs suggests they play a role in endolymph resorption.

Mitochondria-rich cells express SLC26A4 (also known as pendrin), an anion exchanger encoded by the *SLC26A4* gene. Mutations of *SLC26A4* are the most common cause of EVA and the first or second most common cause of childhood deafness worldwide (*Park et al., 2003*). The mouse model deficient in SLC26A4 ($Slc26a4^{\Delta/\Delta}$) has profound hearing loss, vestibular dysfunction, and massively enlarged endolymphatic spaces throughout the entire inner ear (*Everett et al., 2001*). Our previous studies of transgenic mouse models indicate that *Slc26a4* expression is required from embryonic day 16.5 (E16.5) to postnatal day 2 (P2) in the endolymphatic sac but, remarkably, not the cochlea for the development of normal hearing (*Li et al., 2013b*; *Choi et al., 2011*). Mutations of other genes that are expressed in MRCs also cause EVA in humans, mouse models, or both including *Atp6v0a4*, *Atp6v1b1*, and *Foxi1* (*Hulander et al., 2003*; *Lorente-Cánovas et al., 2013*). *Atp6v0a4* and *Atp6v1b1* encode subunits of a vacuolar-type $H^+$-ATPase (v-ATPase) expressed in the apical membrane of MRCs (*Dou et al., 2003*; *Dou et al., 2004*; *Vidarsson et al., 2009*). *Foxi1* encodes a forkhead transcriptional factor that regulates expression of the genes encoding SLC26A4, specific subunits of the v-ATPase, and the bicarbonate transporter SLC4A9 (AE4) (*Raft et al., 2014*; *Hulander et al., 2003*; *Vidarsson et al., 2009*; *Kurth et al., 2006*). MRCs are thus one of a family of cell types known as FORE (forkhead-related) cells that include intercalated cells of the renal collecting duct as well as narrow and clear cells of the epididymidis (*Vidarsson et al., 2009*). Other known expression markers of MRCs in the endolymphatic sac are carbonic anhydrase 2 (encoded by *Car2*), cochlin (*Coch*), and jagged 1 (*Jag1*) (*Hulander et al., 2003*; *Dou et al., 2004*). Although gene and protein expression profiles have been obtained for the endolymphatic sac of the rat and human (*Kim et al., 2015*; *Møller et al., 2015a*; *Møller et al., 2015b*), these studies did not differentiate between MRCs and RRCs or investigate the possibility that additional types of epithelial cells are present.

The purpose of our present study was to gain insight into the functional, molecular and cellular architecture of the endolymphatic sac and to identify the components of the physiologic-developmental pathway that is disrupted in EVA. We show that the endolymphatic sac absorbs fluid in a SLC26A4-dependent fashion. We use single-cell RNA-seq and gene array analyses to elucidate the physiology, cellular architecture and development of the endolymphatic sac, as well as to generate a model of the molecular mechanism for NaCl resorption during morphogenesis. We propose that a disruption of this mechanism in MRCs is the root cause of EVA and hearing loss.

## Results

### Fluid absorption by endolymphatic sacs in vitro

We sought to obtain direct evidence that the embryonic endolymphatic sac is engaged in absorption of fluid from its lumen. Endolymphatic sacs were isolated from E14.5 $Slc26a4^{\Delta/+}$ and $Slc26a4^{\Delta/\Delta}$ mice of either sex, filled with a solution containing the fluorescent dye seminaphtarhodafluor (SNARF-1) and maintained in organ culture (*Figure 1*). Embryonic day 14.5 was chosen for functional studies because it is the earliest age at which a difference in the fluid volume of the inner ear was detected between $Slc26a4^{\Delta/+}$ and $Slc26a4^{\Delta/\Delta}$ mice. Furthermore, the attachment between epithelial cells and connective tissue could be separated at this age without perforation of the epithelial monolayer (*Kim and Wangemann, 2010*). We obtained the initial volume and luminal surface area at time zero and repeated volume measurements in semi-hourly intervals to quantify the rate of fluid absorption. Rates of fluid absorption were normalized to the luminal surface area determined at time zero. Three groups of experiments were performed.

In the first group of experiments, we evaluated fluid absorption in E14.5 $Slc26a4^{\Delta/+}$ sacs with and without abluminal administration of ouabain and in E14.5 $Slc26a4^{\Delta/\Delta}$ mice. The rate of fluid absorption was higher in endolymphatic sacs from $Slc26a4^{\Delta/+}$ mice ($12.3 \pm 2.6$ nl x hr$^{-1}$ x mm$^{-2}$, n = 26) than in endolymphatic sacs from $Slc26a4^{\Delta/\Delta}$ mice ($6.0 \pm 1.6$ nl x hr$^{-1}$ x mm$^{-2}$, n = 12, One-way ANOVA: p<0.001, Cohens's d = 3.0; *Figure 1—source data 1*) or in endolymphatic sacs from $Slc26a4^{\Delta/+}$ mice in the presence of ouabain, a blocker of Na$^+$/K$^+$ ATPase (1 mM, $5.3 \pm 1.8$ nl x hr$^{-1}$ x mm$^{-2}$, n = 7, One-way ANOVA: p<0.001, Cohen's d = 3.2). Expression of ATP1A1, the alpha 1 sub-unit of Na$^+$/K$^+$ ATPase, was detected in basolateral membranes of all epithelial cells in the endolymphatic sac and duct (*Figure 1E–G*).

In the second group of experiments, we evaluated fluid absorption in the presence of benzamil, bafilomycin, or S3226. Bafilomycin is an inhibitor of the v-ATPase (*Dröse and Altendorf, 1997*). Benzamil is an inhibitor of the epithelial Na$^+$ channel ENaC (*Cuthbert and Edwardson, 1979*), and S3226 is an inhibitor of the Na$^+$/H$^+$ exchanger NHE3 (*Schwark et al., 1998*). Preparation of solutions containing bafilomycin, benzamil or S3226 required a vehicle for which we used dimethyl sulfoxide (DMSO). Neither the addition of benzamil (10 µM or 100 µM, one-way ANOVA: p=1), the addition of bafilomycin A1 (10 nM or 10 µM, one-way ANOVA: p=1), nor the addition of S3226 (20 µM, one-way ANOVA: p=1) to the luminal fluid had a significant effect on the rate of fluid absorption in endolymphatic sacs of E14.5 $Slc26a4^{\Delta/+}$ mice.

In the third group of experiments, we evaluated fluid absorption in the presence of gadolinium which is an inhibitor of non-selective cation channels (*Hamill and McBride, 1996*; *Nakazawa et al., 1997*). The rate of fluid absorption was higher in endolymphatic sacs from $Slc26a4^{\Delta/+}$ mice in the absence ($12.6 \pm 2.7$ nl x hr$^{-1}$ x mm$^{-2}$, n = 6) than in the presence of gadolinium ($8.1 \pm 1.5$ nl x hr$^{-1}$ x mm$^{-2}$, n = 11, One-way ANOVA: p=0.008, Cohens's d = 2.2; *Figure 1—source data 1*).

Taken together, these observations demonstrate that fluid absorption in the embryonic endolymphatic sac at E14.5 depends on SLC26A4, Na$^+$/K$^+$ ATPase, and a gadolinium-sensitive pathway. Neither v-ATPase, ENaC Na$^+$ channels, nor the Na$^+$/H$^+$ exchanger NHE3 are required at this age.

### Transcriptomes of single cells from postnatal endolymphatic sac epithelium

We next sought to sequence transcriptomes of single cells isolated from endolymphatic sac epithelia of C57BL/6J mice at postnatal day 5 (P5). We selected only clearly-identified single cells, leaving 82 P5 epithelial cells for RNA-seq analysis (Materials and methods, *Figure 2—source data 1*). We conducted principal component analysis (PCA) of the RNA-seq data using the 16,532 genes expressed (normalized transcripts per million (nTPM) >1) in more than two cells and with a cell-to-cell coefficient of variation >0.5. The first principal component (PC1) separated cells into two distinct populations in an unbiased fashion (*Figure 2A*). The canonical MRC markers (*Slc26a4, Foxi1, Atp6v0a4, Atp6v1b1, Jag1,* and *Coch)* showed high positive correlation with PC1, whereas *Notch1*, which is the only known gene expression marker of embryonic non-MRCs (*Raft et al., 2014*), showed a negative correlation with PC1 (*Figure 2B*). Hierarchical clustering analysis using the 300 genes with the highest positive or negative correlation with PC1 exhibited clear segregation of each of the 82 cells into one of two primary clusters, and segregation of each of the 300 genes into one of two primary

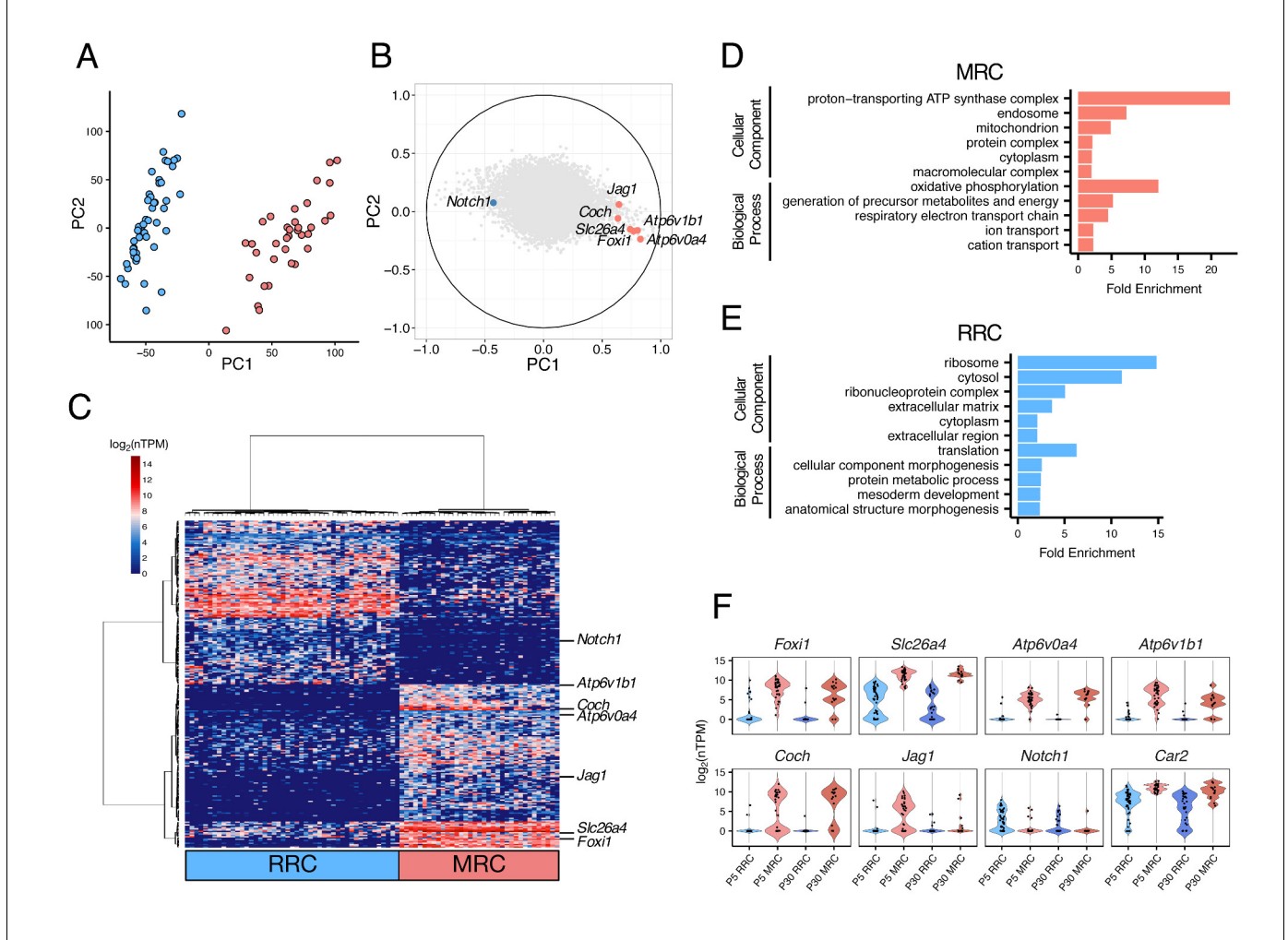

**Figure 2.** Single-cell RNA-seq enables unbiased clustering of postnatal endolymphatic sac epithelial cells. (A) Plot of single-cell transcriptomes of 82 P5 endolymphatic sac epithelial cells (captured on three C1 IFCs) projected onto the first two principal components (PCs) calculated by PCA using all expressed genes. Cells clearly separated into two groups. (B) Correlation circle map on PC1 and PC2. Each point corresponds to a gene and the known cell markers are highlighted in color. These results suggest that PC1 mainly contributes separation of MRCs and RRCs. (C) Hierarchical clustering of 82 P5 cells (x-axis) using the top 300 genes (y-axis) that are highly correlated with PC1 positively or negatively. Two distinct clusters of cells were identified and assigned into MRCs and RRCs based upon the expression of the known markers. (D–E) Functional enrichment analysis of gene ontology (GO) terms was performed for 594 differentially expressed genes in MRCs (MRC genes) and 426 differentially expressed genes in RRCs (RRC genes) at P5 (false discovery rate: FDR < 0.05, specificity score >0.5, *Figure 2—source data 2*) from the Gene Ontology Consortium website. Only annotations of PANTHER GO-Slim dataset with the Bonferroni corrected p-value of <0.05 and fold-enrichment >2 are shown. (F) Violin plots of canonical MRC and RRC genes. Four known EVA genes and other previously reported marker genes were differentially expressed between P5 MRCs and P5 RRCs, or between P30 MRCs and P30 RRCs (*Figure 2—figure supplement 1*). Although *Slc26a4* was significantly highly expressed in P5 and P30 MRCs, a subset of RRCs express *Slc26a4* at low levels (*Figure 2—figure supplement 2*). *Jag1* and *Notch1* were not differentially expressed at P30. Black dots show the expression level for each cell.

DOI: https://doi.org/10.7554/eLife.26851.005

The following source data and figure supplements are available for figure 2:

**Source data 1.** Summary of numbers of cells captured and sequenced.

DOI: https://doi.org/10.7554/eLife.26851.008

**Source data 2.** Cell-type specific genes identified by differential expression analysis.

DOI: https://doi.org/10.7554/eLife.26851.009

**Source data 3.** List of TaqMan® gene expression assays.

DOI: https://doi.org/10.7554/eLife.26851.010

**Figure supplement 1.** Unbiased clustering of P30 endolymphatic sac epithelial cells.

DOI: https://doi.org/10.7554/eLife.26851.006

*Figure 2 continued on next page*

*Figure 2 continued*

**Figure supplement 2.** A subset of RRCs express pendrin at low levels.

DOI: https://doi.org/10.7554/eLife.26851.007

clusters (*Figure 2C*). Based upon expression patterns of the canonical cell markers, the two cell clusters were assigned to P5 MRCs and P5 RRCs.

In order to identify genes whose expression was unique to either cell type, we tested for differential expression between the 37 MRCs and the 45 RRCs, as reported previously (*Burns et al., 2015*). We identified 433 genes whose expression was specific to RRCs ('RRC genes') and 611 genes whose expression was specific to MRCs ('MRC genes') according to criteria of significant differential expression (false discovery rate (FDR) < 0.05 and specificity >0.5) (*Figure 2—source data 2*).

In order to identify functional pathways associated with the two differentially expressed gene sets, we performed gene ontology (GO) enrichment analysis. We found certain GO terms more frequently represented in one or the other gene set (Bonferroni corrected p-value<0.05) (*Figure 2D,E*). Gene ontology terms that were enriched for P5 MRC genes included 'mitochondrion' (GO:0005739) and other terms implicating mitochondrial function such as 'generation of precursor metabolites and energy' (GO:0006091) and 'respiratory electron transport chain' (GO:0022904). Postnatal day 5 RRC genes were significantly associated with the GO terms 'ribosome' (GO:0005840) and 'translation' (GO:0006412). These findings indicated that our unbiased clustering and subsequent differential expression analysis can effectively identify different populations of endolymphatic sac epithelial cells as well as cell-specific genes for each population.

To clarify how the cell populations and their transcriptomes mature with age, we sequenced single cells of P30 endolymphatic sac epithelia. The results were similar to those for P5 cells. Principal component analysis and hierarchical clustering identified two cell populations among 46 P30 cells, which were readily assigned to MRCs or RRCs on the basis of canonical and putative population-specific genes identified by the P5 analysis (*Figure 2—figure supplement 1A,B*). This result indicates that there is no qualitative difference in endolymphatic sac cellular composition between P5 and P30. The subsequent differential expression analysis (FDR < 0.05, specificity >0.5) identified 316 MRC-specific genes and 222 RRC-specific genes at P30 (*Figure 2—source data 2*). To focus on genes more specific to P30 MRCs and RRCs, we combined two datasets at P5 and P30 and performed differential expression analysis across four cell groups (P5 MRCs, P5 RRCs, P30 MRCs, and P30 RRCs). We identified four MRC-specific genes (*Chchd10, Coch, Atp6v1g3,* and *Tnrc6b*) and 9 RRC-specific genes (*Slpi, Serping1, Cp, Clu, Sbsn, Trf, Sostdc1, Lcn2,* and *Rarres2*) whose expression was significantly increasing at P30 in comparison to P5 (FDR < 0.05, specificity >0.65) (*Figure 2—figure supplement 1C,D*). The nine genes whose expression was specific to P30 RRCs were associated with GO terms such as 'iron ion homeostasis' (GO:0055076; p-value $2.14 \times 10^{-6}$; *Cp, Trf, Lcn2*), 'defense response' (GO:0006952; p-value $2.54 \times 10^{-6}$; *Lcn2, Rarres2, Slpi, Serping1, Trf*), or 'innate immune response' (GO:0045087, p-value $7.22 \times 10^{-4}$; *Lcn2, Slpi, Serping1*), implying that mature RRCs might contribute to immunity of the inner ear.

Interestingly, our single-cell RNA-seq data indicate that approximately one half of P5 and P30 RRCs express *Slc26a4* at a low level (*Figure 2F*). The mean expression levels of *Slc26a4* in *Slc26a4*-positive RRCs (>5 $\log_2$ (nTPM)) are 14-to 18-fold lower than those for MRCs. To validate the expression of *Slc26a4* in the non-MRC cell population, we performed single-cell qPCR of epithelial cells isolated from C57BL/6J P5 endolymphatic sacs. Consistent with the result of single cell RNA-seq, we found that some cells expressed *Slc26a4* but not other MRC-specific genes (*Figure 2—figure supplement 2A*). Immunohistochemistry (IHC) of P5 endolymphatic sacs verified the presence of SLC26A4 in ATP6V1B1-negative cells (*Figure 2—figure supplement 2B*) and FOXI1-negative cells (*Figure 2—figure supplement 2C*). These findings indicate that a subset of RRCs express SLC26A4 at low levels.

## Transcriptomes of single cells from pre-natal endolymphatic sac epithelium

Next, we performed the same analyses for 44 E12.5 cells and 41 E16.5 cells. Principal component analysis did not identify any distinct clusters of E12.5 cells (*Figure 3—figure supplement 1*),

whereas E16.5 cells were clustered into three populations (*Figure 3A,B*). The 100 genes with the highest correlation with PC1 were divided into three clusters by hierarchical clustering. The first gene cluster (*Figure 3B*, bottom rows) included canonical MRC markers (*Slc26a4*, *Atp6v1b1*, and *Atp6v0a4*) and other MRC genes identified by analyses of P5 and P30 samples, so we assigned the corresponding cell population as E16.5 MRCs. The second gene cluster (*Figure 3B*, middle rows) was comprised of nine genes including seven RRC genes at P5 (*Vim*, *Mme*, *Nsg1*, *Prelp*, *Oraov1*, *Pdlim1*, *Nox3*, *Ctnnd2*, and *Psat1*). The third gene cluster included 29 genes associated with the GO term 'cell cycle' (GO:0007049), indicating the gene cluster corresponds to proliferating cells. Thus, we designated the remaining cell populations as E16.5 progenitor cells (ProgCs) and E16.5 proliferating cells (ProlCs), respectively. The ProgCs and ProlCs both showed expression of (RRC) genes in cluster 2. Differential expression testing between MRCs and ProgCs identified 975 E16.5 MRC genes and 278 E16.5 ProgC genes (FDR < 0.05, specificity >0.5) (*Figure 2—source data 2*).

At E16.5, *Foxi1* expression appeared to be less specific to MRCs than it was at P5 or P30. In addition to high expression of *Foxi1* in all MRCs, a subset of ProgCs and ProlCs also showed high expression levels of *Foxi1* (*Figure 3B*). In addition, we observed that some ProgCs and ProlCs express other MRC genes, although the expression levels are lower than in MRCs. From these observations, we hypothesized that ProgCs and ProlCs expressing *Foxi1* or other MRC genes represent a transitional state of progenitor cells that are proliferating and differentiating into MRCs.

## Cell trajectory analysis of transcriptomes in developing endolymphatic sac

We performed PCA on all 213 cells from E12.5 to P30 (*Figure 4A,B*). PC2 clearly separates MRCs from other cells at all time points (*Figure 4A*). PC3 segregates E12.5 cells and PC4 segregates E16.5 ProlCs from other cells. Then we performed hierarchical clustering using the top 25 genes positively and negatively correlated with PC2, the top 25 genes positively and negatively correlated with PC3, and the top 25 genes negatively correlated with PC4. Two genes were present in two of these categories, yielding a total of 123 genes for cluster analysis (*Figure 4C*). Each of the 123 genes were

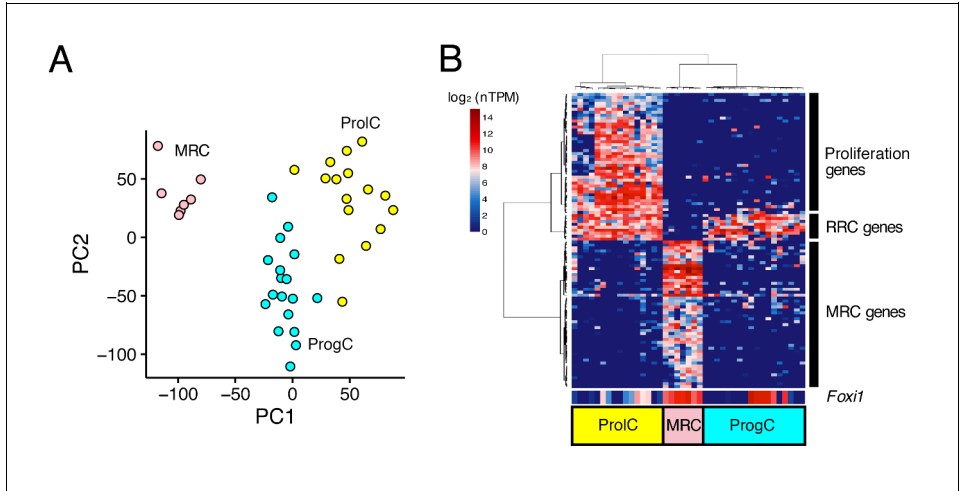

**Figure 3.** Unbiased clustering of E16.5 endolymphatic sac epithelial cells. (**A–B**) Plots show PCA (**A**) and hierarchical clustering (**B**) of 41 E16.5 cells (captured on two C1 IFCs). Cells are clustered into three groups, which are color-coded and labeled as MRCs (pink), progenitor cells (ProgCs, light blue), and proliferating cells (ProlCs, yellow). Seven MRCs were segregated from other cells by PC1. The top 100 genes with the highest correlation to PC1 were used for hierarchical clustering. The genes are clustered into three groups, which are assigned to MRC genes, RRC genes, and proliferation genes (29 out of 39 genes are associated with the GO term 'cell cycle' (GO:0007049)). *Foxi1* is not specific to MRCs at E16.5.

DOI: https://doi.org/10.7554/eLife.26851.011

The following figure supplement is available for figure 3:

**Figure supplement 1.** Principal component analysis of 44 E12.5 cells.

DOI: https://doi.org/10.7554/eLife.26851.012

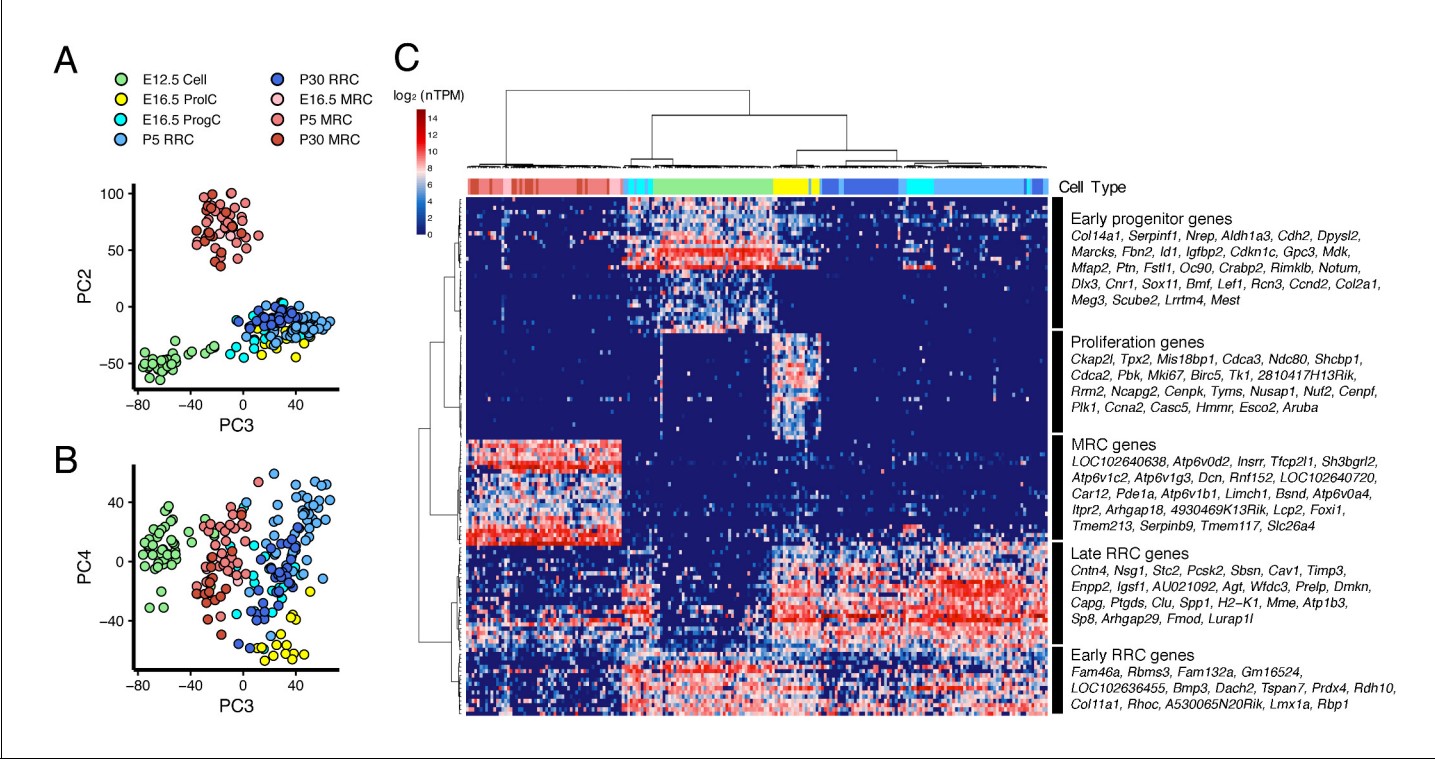

**Figure 4.** Cross-age transcriptome analysis resolves cell differentiation pathway of endolymphatic sac. (A–B) PCA plots of 213 endolymphatic sac epithelial cells onto PC2, PC3 and PC4. Each cell has already been annotated and color-coded by prior analyses (*Figures 2* and *3*, *Figure 2—figure supplement 1*). PC2, PC3, and PC4 contribute to segregation of MRCs, E12.5 cells and ProlCs from others, respectively. (C) Hierarchical clustering of the 213 cells using the 123 genes with positive or negative correlation to PC2 or PC3, or with negative correlation to PC4. The same color-codes as those used in (A–B) were shown (top color bar). Four MRC genes, LOC192640638, LOC102640720, *Lcp2*, and 4930469K13Rik, are located adjacent to *Foxi1* on chromosome 11 (*Figure 4—figure supplement 1*).

DOI: https://doi.org/10.7554/eLife.26851.013

The following figure supplement is available for figure 4:

**Figure supplement 1.** Potential MRC-specific lncRNAs located adjacent to *Foxi1*.

DOI: https://doi.org/10.7554/eLife.26851.014

clustered into one of five groups: (1) genes specific to MRCs (MRC genes); (2) genes mainly specific to E12.5 cells (early progenitor genes); (3) genes specific to all E16.5 ProlCs, two P5 RRCs and one E12.5 cell (proliferation genes); (4) genes specific to E16.5 ProgCs and ProlCs, P5 RRCs and P30 RRCs (late RRC genes); and (5) genes specific to E12.5 cells, E16.5 ProgCs/ProlCs, and P5 RRCs (early RRC genes). Whereas MRCs at all time points formed a single cluster, E12.5 cells, E16.5 ProgCs, E16.5 ProlCs, P5 RRCs, and P30 RRCs formed another large cluster partially sharing expression profiles of early progenitor genes, early RRC genes, and late RRC genes. This result suggests these latter cell groups constitute the main pathway of differentiation in endolymphatic sac epithelium. In addition, only a few cells at E12.5 and P5 showed gene expression profiles associated with proliferation, indicating proliferation may occur most actively at E16.5. We tested this by counting 5-ethynyl-2′-deoxyuridine (EdU)-positive cells, as a measure of proliferation, in whole mount samples of endolymphatic sac epithelium from E12.5 to P5 (*Figure 5A*). Our hypothesis was supported by a maximal percentage of EdU-positive cells detected at E16.5 (*Figure 5B*).

We performed computational single-cell trajectory analysis using Monocle 2 (*Trapnell et al., 2014*). Monocle 2 ordered cells at E12.5, E16.5 and P5 in an unsupervised fashion and then reconstructed a branched trajectory, beginning at E12.5 cells, passing through E16.5 ProlCs and ProgCs, and ending at two outcomes, P5 MRCs or P5 RRCs (*Figure 6A*). In the trajectory plot, six E16.5 ProgCs, one E16.5 ProlC, and two P5 RRCs were located along the MRC branch, indicating these cells are likely to be MRCs or progenitor cells for MRCs. mRNA levels of representative genes were

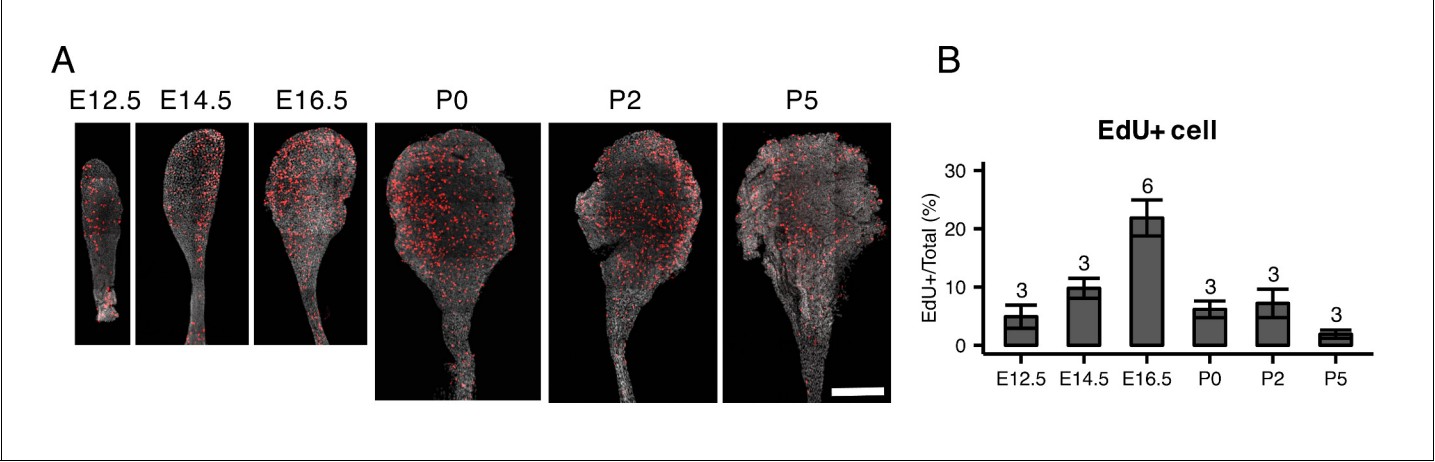

**Figure 5.** EdU staining of whole mount endolymphatic sacs during development. (**A**) Average intensity projection of whole mount endolymphatic sacs from E12.5 to P5 labeled by EdU (red) and DAPI (grey). Original images were captured as a tiled z-stack with confocal microscopy. (**B**) The percentages of EdU-labeled cells in the endolymphatic sac epithelia. Values are expressed as mean ± SD and the total number of embryo analyzed is indicated above each bar plot.
DOI: https://doi.org/10.7554/eLife.26851.015

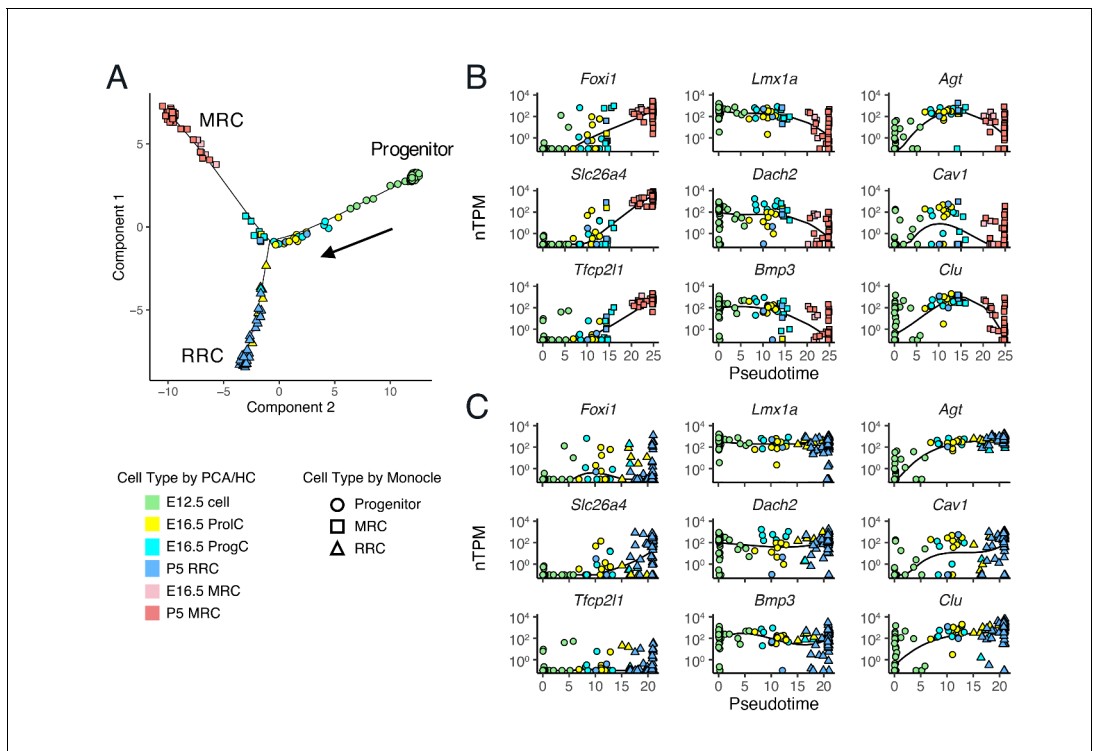

**Figure 6.** Single-cell trajectory analysis of E12.5 to P5 cells using Monocle 2. (**A**) The plot shows single-cell expression profiles of differentially expressed genes (FDR < 0.01) across time points in a two-dimensional component space. The unsupervised ordering method reconstructs a branched trajectory from progenitors to MRCs (squares) or RRCs (triangles). Cell types assigned by PCA and hierarchical clustering (HC) are color-coded. Note that a subset of E16.5 ProgCs, E16.5 ProlCs, and P5 RRCs are clustered into the branch of MRCs. (**B–C**) Expression levels of representative MRC genes (*Foxi1*, *Slc26a4*, *Tfcp2l1*), early RRC genes (*Lmx1a*, *Dach2*, *Bmp3*), and late RRC genes (*Agt*, *Cav1*, *Clu*) in cells are ordered along pseudo-time. Two branches are shown separately: (**B**), progenitor to MRC; and (**C**), progenitor to RRC. Kinetic curves of two branches are shown as solid lines.
DOI: https://doi.org/10.7554/eLife.26851.016

plotted against pseudotime (*Figure 6B,C*). Along the MRC trajectory, mRNA levels of MRC genes such as *Foxi1*, *Slc26a4*, and *Tfcp2l1* are increased, mRNA levels of early RRC genes such as *Lmx1a*, *Dach2*, and *Bmp3* are decreased, and late RRC genes such as *Agt*, *Cav1*, and *Clu* show transient changes in mRNA levels. Moreover, the RRC trajectory shows constant mRNA expression patterns of early RRC genes and increasing mRNA levels of late RRC genes. These results based on the trajectory analysis support the hypothesis that MRCs are derived from the main differentiation pathway between early embryonic cells and postnatal RRCs.

Principal component and differential expression analyses identified four MRC genes, LOC102640638, LOC102640720, *Lcp2*, and 4930469K13Rik, as highly specific to MRCs (*Figure 4—figure supplement 1A*). Interestingly, these genes are located adjacent to *Foxi1* on chromosome 11 (*Figure 4—figure supplement 1B*). Inspection of the RNA-seq read alignments and predicted splicing junctions suggests the four putative genes actually comprise two genes encoding transcripts that are not annotated in the reference genome (GRCm38/mm10) (*Figure 4—figure supplement 1C*). The transcripts are predicted to be long non-coding RNAs (lncRNAs) because they lack open-reading frames encoding more than 100 amino acids. These lncRNAs may act in some manner with *Foxi1* to regulate differentiation into MRCs.

## Ion transporters and channels in endolymphatic sac epithelium

Since the endolymphatic sac has been implicated in fluid, pH and ionic homeostasis during development, we sought to identify candidate genes and cells mediating these processes. In order to generate a refined list of MRC genes and RRC genes across time points, we filtered in genes that were differentially expressed at two or more continuous time points (E16.5 and P5, P5 and P30, or all ages). Three-hundred-sixty MRC genes and 222 RRC genes were thus retained for further analysis. Among the two gene-sets, we identified a total of 55 genes annotated as 'ion transmembrane transporter activity' (GO:0015075) (*Table 1*). Forty-seven were MRC genes and eight were RRC genes (*Table 1*). The MRC genes include known marker genes (*Slc26a4*, *Atp6v1b1*, *Atp6v0a4*), tissue-specific v-ATPase subunit genes (*Atp6v0d2*, *Atp6v1c2*, *Atp6v1g3*), chloride channel genes (*Clcnkb*, *Bsnd*, *Cftr*), an anion exchanger gene (*Slc4a9*), a BK channel α subunit gene (*Kcnma1*), a sodium/phosphate transporter gene (*Slc34a2*), and an epithelial sodium channel α subunit gene (*Scnn1a*) (*Figure 7A*). Expression of a subset of these markers was verified by immunohistochemistry (*Figure 7B,C*).

## Gene array analysis of the endolymphatic sac in *Slc26a4*$^{\Delta/+}$ and *Slc26a4*$^{\Delta/\Delta}$ mice

One limitation of the single-cell RNA-seq technique is that the process of single-cell dissociation may alter the results. Rapid extraction of RNA from intact embryonic endolymphatic sac for microarray analysis does not require enzymatic digestion. To verify expression of ion channel/transporter genes, and to examine what genes are affected by *Slc26a4* disruption, we analyzed gene expression profiles of undissociated, whole endolymphatic sac tissues in *Slc26a4*$^{\Delta/+}$ and *Slc26a4*$^{\Delta/\Delta}$ mice at E13.5, E14.5, E16.5 and E17.5 using Affymetrix microarrays (*Figure 8—figure supplement 1A,B*). We used a regression-based approach to identify genes with significant temporal expression changes and significant differences between genotypes. Signal intensities for 680 probes (525 genes) significantly increased during the time interval in *Slc26a4*$^{\Delta/+}$, *Slc26a4*$^{\Delta/\Delta}$, or both (*Figure 8A* and *Figure 8—source data 2*, Clusters 1, 2, 3, and 6). These genes include 38 'ion transmembrane transporter activity' genes (GO:0015075): *Atp1a1*, *Atp1b1*, *Atp1b3*, *Atp2a3*, *Atp6v0a4*, *Atp6v0c*, *Atp6v0d2*, *Atp6v1a*, *Atp6v1b1*, *Atp6v1c1*, *Atp6v1c2*, *Brp44l*, *Bsnd*, *Chrna4*, *Clcnkb*, *Cnnm4*, *Itgav*, *Kcnk5*, *Kcnn4*, *Kcnq1*, *Lrrc8c*, *S100a6*, *Sat1*, *Scnn1a*, *Scnn1b*, *Slc16a1*, *Slc1a1*, *Slc1a3*, *Slc20a2*, *Slc26a4*, *Slc26a7*, *Slc4a7*, *Slc4a9*, *Slc4a11*, *Slc7a11*, *Slco2a1*, *Ttyh1*, *Tusc3*, *Uqcrfs1*, and *Uqcrh*. Other genes include *Car2* and *Car12* (encoding carbonic anhydrase 2 and 12) and *Foxi1*. Among them, 71 probes (64 genes) are significantly downregulated in *Slc26a4*$^{\Delta/\Delta}$ mice (Cluster 1), including *Atp1b1*, *Atp6v0d2*, *Atp6v1c1*, *Bsnd*, *Chrna4*, *Clcnkb*, *Slc1a1*, *Slc26a4*, *Uqcrfs1*, and *Uqcrh1*. The decreasing levels of *Clcnkb* and *Bsnd* mRNA may reflect a regulatory relationship between *Slc26a4* and Cl⁻ channel genes.

We examined the microarray data for gene expression patterns that were consistent with several known renal mechanisms for NaCl reabsorption. First, the model of the thick ascending limb and the

**Table 1.** Differentially expressed genes associated with 'ion transmembrane transporter activity'

| Cell type | Category | Gene symbol | Protein/Subunit |
|---|---|---|---|
| MRC | V-ATPase (V0 domain) | Atp6v0a4 | a4 (kidney, epididymis) |
| | | Atp6v0b | b (c'') (ubiquitous) |
| | | Atp6v0c | c (ubiquitous) |
| | | Atp6v0d2 | d2 (kidney, epididymis) |
| | V-ATPase (V1 domain) | Atp6v1a | A (ubiquitous) |
| | | Atp6v1b1 | B1 (kidney, epididymis) |
| | | Atp6v1c2 | C2 (lung, kidney, epididymis) |
| | | Atp6v1e1 | E1 (ubiquitous) |
| | | Atp6v1f | F (ubiquitous) |
| | | Atp6v1g3 | G3 (kidney, epididymis) |
| | | Atp6v1h | H (ubiquitous) |
| | $Cl^-/HCO_3^-$ exchanger | Slc26a4 | Pendrin |
| | $Na^+/HCO_3^-$ cotransporter, $Cl^-/HCO_3^-$ exchanger | Slc4a9 | AE4 |
| | $Na^+$/phosphate cotransporter | Slc34a2 | NPT2B |
| | $Na^+$ channel | Scnn1a | ENaC α subunit |
| | $K^+$ channel | Kcnma1 | MaxiK/BK |
| | $Ca^{2+}$ channel | Itpr2 | ITPR2 |
| | $Cl^-$ channel | Clcnkb | CLC-Kb |
| | | Bsnd | Barttin |
| | | Cftr | CFTR |
| | Voltage-gated anion channel | Vdac1 | VDAC1 (porin) |
| | | Vdac2 | VDAC2 |
| | Integrin | Itgav | Integrin α-V |
| | $Na^+$/amino acid symporter | Slc1a1 | EAAT3 |
| | L-amino acid transporter | Slc43a2 | LAT4 |
| | | Serinc2 | Serine incorporator 2 |
| | | Serinc4 | Serine incorporator 4 |
| | Pyruvate transporter | Mpc1 | Mitochondrial pyruvate carrier 1 |
| | ATP Synthase (F1 domain) | Atp5a1 | α |
| | | Atp5b | β |
| | | Atp5o | O |
| | ATP Synthase (F0 domain) | Atp5f1 | B1 |
| | | Atp5g1 | C1 |
| | | Atp5g3 | C3 |
| | | Atp5h | D |
| | | Atp5j | F6 |
| | | Atp5l | G |
| | Cytochrome c oxidase | Cox5a | 5A |
| | | Cox6c | 6C |
| | | Cox7b | 7B |
| | | Cox7c | 7C |
| | | Cox8a | 8A |

*Table 1 continued on next page*

Table 1 continued

| Cell type | Category | Gene symbol | Protein/Subunit |
|---|---|---|---|
| | Ubiquinol-cytochrome c reductase | Uqcr10 | Subunit 9 |
| | | Uqcr11 | Subunit 10 |
| | | Uqcrb | Subunit 7 |
| | | Uqcrfs1 | Subunit Rieske |
| | | Uqcrh | Subunit 6 |
| RRC | Na$^+$/K$^+$ ATPase | Atp1b3 | Na$^+$/K$^+$ ATPase subunit β3 |
| | Ca$^{2+}$ channel | Cacna1g | Cav3.1 |
| | Cation channel | Trpm3 | TRPM3 |
| | Cl$^-$ channel | Fxyd3 | FXYD3 (MAT-8) |
| | Volume-sensitive anion channel | Lrrc8c | LRRC8C |
| | Gap junction | Gja1 | GJA1 |
| | Na$^+$/amino acid sympoter | Slc1a3 | EAAT1 |
| | H$^+$/amino acid symporter | Slc36a3 | PAT3 |

DOI: https://doi.org/10.7554/eLife.26851.018

model of the renal connecting tubule were rejected on the basis that the embryonic endolymphatic sac lacks expression of mRNAs encoding the Na$^+$/2Cl$^-$/K$^+$ co-transporter SLC12A1 and the thiazide-sensitive Na$^+$/Cl$^-$ co-transporter SLC12A3. In contrast, NaCl absorption in the cortical collecting duct was identified as a reasonable model describing NaCl absorption in the embryonic endolymphatic sac (*Wall and Lazo-Fernandez, 2015*). Critical genes shared between the cortical collecting duct and the embryonic endolymphatic sac include *Slc26a4*, and genes encoding subunits of the v-ATPase (*Atp6v1b1, Atp6v1c2, Atp6v0a4, Atp6v0d2*), carbonic anhydrase 2 and 12 (*Car2, Car12*), the transcription factor *Foxi1*, Cl$^-$ channel subunits (*Cftr, Clcnkb,* and *Bsnd*), the Cl$^-$/HCO$_3^-$ exchanger *Slc4a9*, ENaC subunits (*Scnn1a* and *Scnn1b*), a K$^+$ channel subunit (*Kcnq1*) and a Na$^+$/K$^+$ ATPase subunit (*Atp1a1*) (*Figure 8B*). According to our RNA-seq data, all these genes except *Scnn1b, Kcnq1,* and *Atp1a1* are specifically expressed in MRCs. In contrast, expression of each of these genes is distributed between α- and β-intercalated cells and principal cells in the renal collecting duct.

Immunohistochemistry demonstrated there was no difference of FOXI1-positive cell ratios in endolymphatic sacs between *Slc26a4$^{\Delta/+}$* and *Slc26a4$^{\Delta/\Delta}$* mice at P0 (*Figure 8B–F*). This indicates that the difference in gene expression levels between *Slc26a4$^{\Delta/+}$* and *Slc26a4$^{\Delta/\Delta}$* sacs results from changes of expression at a cellular level, not changes in the number of MRCs.

To assess the validity of the gene array and single-cell RNA-seq data, we compared gene array data of E16.5 *Slc26a4$^{\Delta/+}$* endolymphatic sacs with single-cell RNA-seq data of E16.5 wild-type endolymphatic sacs. (We did not have gene array data for wild-type sacs or RNA-seq data for *Slc26a4$^{\Delta/+}$* sacs to compare identical genotypes.) There were 13,338 genes (26,132 probes) detected by gene array and 16,807 genes detected by single-cell RNA-seq, respectively. The expression of 11,027 genes present in both datasets was compared. Data derived from the two methods show positive correlation for moderately or highly expressed genes (*Figure 8—figure supplement 2A*). Wider dispersion is seen at lower expression levels, which may reflect unreliability of low-end TPM values and high stochastic variation in single-cell RNA-seq data. In addition, we calculated changes of average expression between two time points for each of the two methods. A comparison of the fold-change (expressed as log$_2$) in expression at E16.5 versus E12.5 obtained by single-cell RNA-seq and fold-change in expression at E16.5 versus E13.5 obtained by gene array shows positive correlation (R = 0.38) (*Figure 8—figure supplement 2B*). For representative genes, the direction and magnitude of the expression changes are consistent between the two data sets (*Figure 8—figure supplement 2C*).

Moreover, we assessed gene expression of whole endolymphatic sacs obtained from wild-type mice at ages E12.5 to P90 by RT-qPCR. We compared the temporal expression trends with those

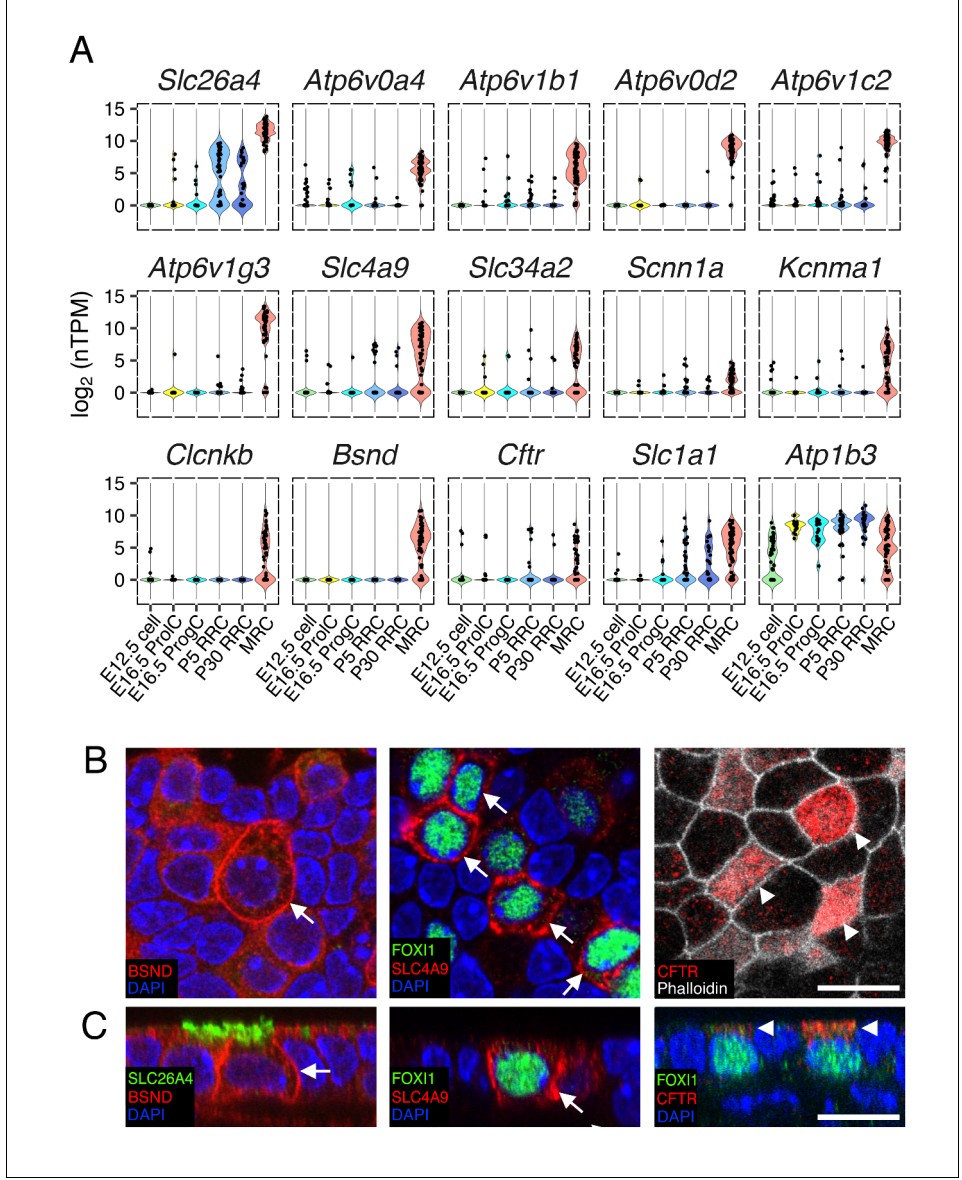

**Figure 7.** Ion transporter/channels in the endolymphatic sac epithelia. (**A**) Violin plots of representative genes differentially expressed in MRCs or RRCs (**Table 1**). (**B–C**) Immunohistochemisty validation of BSND (Barttin), SLC4A9 (AE4), and CFTR. Representative images from z-stacks of whole mount endolymphatic sac epithelia at P5 are shown in (**B**) and reconstructed cross-sections are shown in (**C**). BSND and SLC4A9 localize to basolateral membranes (arrows) and CFTR localizes to apical membranes (arrowheads). Scale bars, 10 μm.
DOI: https://doi.org/10.7554/eLife.26851.017

within the gene array data of $Slc26a4^{\Delta/+}$ endolymphatic sacs, and confirmed the trends were similar to each other (**Figure 8—figure supplement 3**).

## Discussion

Our study provides direct evidence of fluid absorption by the developing endolymphatic sac in a SLC26A4-dependent manner in vitro (**Figure 1B–D**). In conjunction with our previous findings (**Kim and Wangemann, 2010**; **Choi et al., 2011**; **Kim and Wangemann, 2011**), these results confirm our model in which the endolymphatic sac absorbs fluid during morphogenesis of the inner ear.

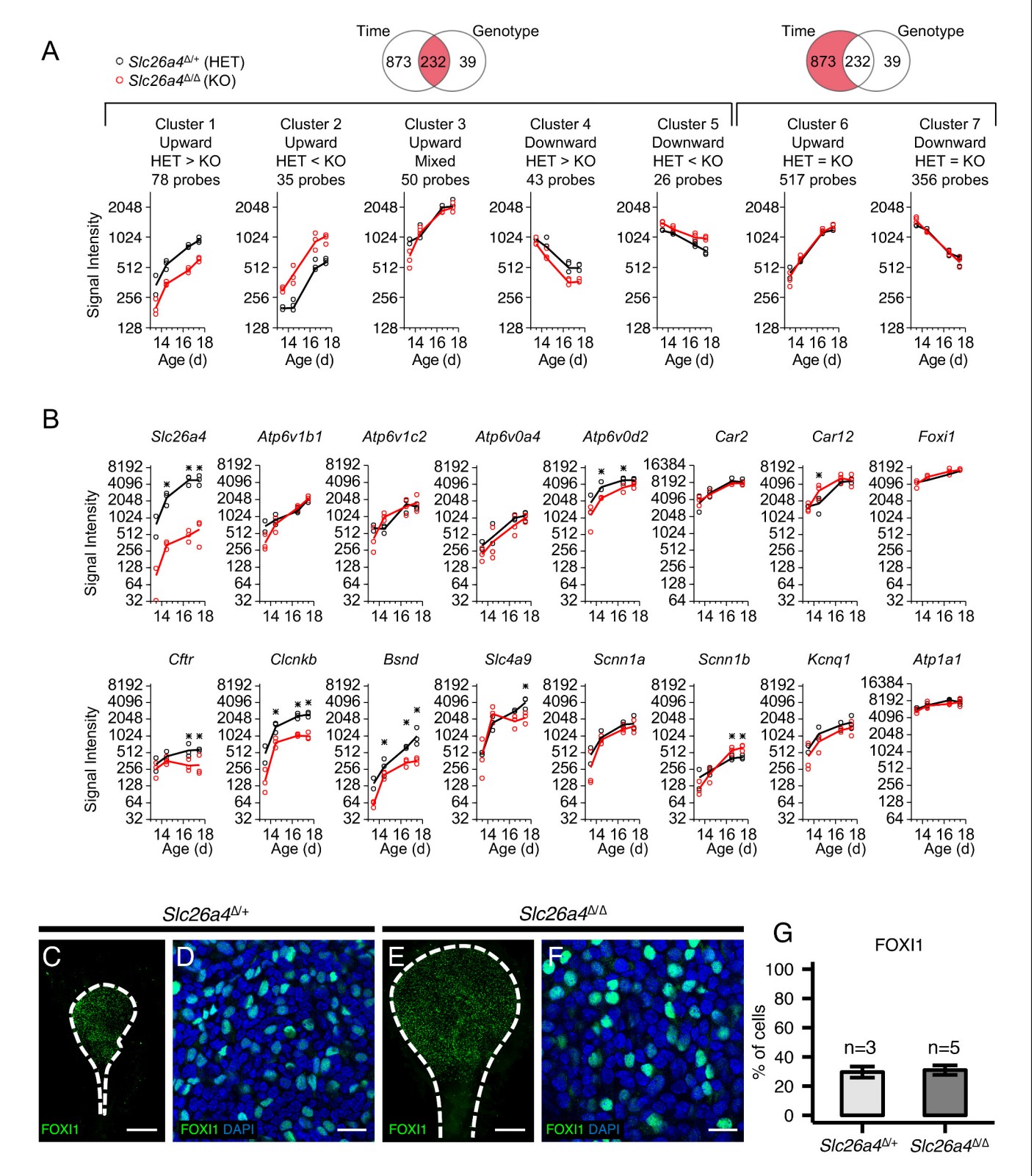

**Figure 8.** Gene expression profiles of embryonic endolymphatic sacs from *Slc26a4*[Δ/+] and *Slc26a4*[Δ/Δ] mice. (**A**) Differential expression analysis and subsequent clustering by temporal expression patterns. Regression-based analysis identified 232 probes with significant differences across time points and between genotypes, and also identified 873 probes only with significant difference across time points. The two significant probe groups are clustered to five and two clusters, respectively, using hierarchical clustering (*Figure 8—source data 2*). Each point (circle or triangle) represents the

*Figure 8 continued on next page*

Figure 8 continued

median signal intensity of multiple probes per biological replicate and each line connects the mean values of replicates. (B) Expression levels of ion transport genes in endolymphatic sacs at ages E13.5, E14.5, E16.5 and E17.5. Each point (black or red circle) represents the signal intensity of a biological replicate and each line represents the mean of the replicates. Expression of all genes increased significantly with age in $Slc26a4^{\Delta/+}$ and $Slc26a4^{\Delta/\Delta}$ mice with the exception of the expression of $Slc26a4$ and $Cftr$ in $Slc26a4^{\Delta/\Delta}$ mice. Significant differences in expression levels between genotypes are indicated with asterisks. (C, E) Maximum intensity projection images of whole mount endolymphatic sacs in P0 $Slc26a4^{\Delta/+}$ and $Slc26a4^{\Delta/\Delta}$ mice labeled by anti-FOXI1 antibodies (green). (D, F) Representative images of magnified Z-stacks of the whole mount endolymphatic sacs. Nuclei were stained with DAPI (blue). (G) Mean percentages of FOXI1$^+$ cells per total cells. Error bars indicate SD. Scale bars, 250 μm (C, E); 20 μm (D, F).

DOI: https://doi.org/10.7554/eLife.26851.019

The following source data and figure supplements are available for figure 8:

**Source data 1.** Quality and quantity of total RNA and quality metrics of gene array chips.

DOI: https://doi.org/10.7554/eLife.26851.023

**Source data 2.** Gene array analysis of embryonic endolymphatic sac in $Slc26a4^{\Delta/+}$ and $Slc26a4^{\Delta/\Delta}$ mice.

DOI: https://doi.org/10.7554/eLife.26851.024

**Figure supplement 1.** Microarray analysis of gene expression of endolymphatic sac.

DOI: https://doi.org/10.7554/eLife.26851.020

**Figure supplement 2.** Comparative analysis of gene array and single-RNA-seq data.

DOI: https://doi.org/10.7554/eLife.26851.021

**Figure supplement 3.** Temporal expression trends of gene array and RT-qPCR data from whole endolymphatic sacs.

DOI: https://doi.org/10.7554/eLife.26851.022

Our evaluation of transcriptomic signatures demonstrates in an unbiased fashion that postnatal endolymphatic sac epithelial cells can be classified into two distinct cell types: mitochondria-rich cells (MRCs) and ribosome-rich cells (RRCs) (*Figure 2C*). Gene ontology analysis of genes that are differentially expressed between these two cell types indicate that these morphologic descriptions are accurate and appropriate (*Figure 2D,E*). Most importantly, our results indicate that MRCs are the primary cells in the developing endolymphatic sac that express ion transport mechanisms necessary for transepithelial fluid absorption. This observation, in combination with the observation that MRCs, not RRCs, express known EVA genes, implicates MRC dysfunction as the initial lesion in the cascade of events leading to EVA and hearing loss.

Based upon functional data and expression profiles of transporters, channels, pumps, and exchangers in MRCs, we have generated a molecular model for fluid absorption in the developing endolymphatic sac (*Figure 9*). The model assumes that fluid absorption is driven by transepithelial NaCl absorption since Na$^+$ and Cl$^-$ are the prevalent solutes in endolymph of the endolymphatic sac (*Li et al., 2013a*; *Couloigner et al., 1998*; *Couloigner et al., 1999*). The onset of NaCl absorption in the embryonic endolymphatic sac between E13.5 and E14.5 coincides with a steep rise in the expression of the genes that are implicated in the model (*Kim and Wangemann, 2010*). These genes include $Slc26a4$, $Scnn1a$ and $Scnn1b$, $Atp6v1b1$ and $Atp6v0a4$, $Slc4a9$, $Clcnkb$ and $Bsnd$ (*Figure 8B*). A central feature of our model is absorption of endolymphatic Cl$^-$ across the apical membrane into the cytosol of MRCs by the Cl$^-$/HCO$_3^-$ exchanger SLC26A4. Na$^+$ is absorbed across the apical membrane via a gadolinium-sensitive pathway and exported across the basolateral membrane by the Na$^+$/K$^+$ ATPase. K$^+$ is recycled in the basolateral membrane through K$^+$ channels that establish a cytosolic-side negative potential across the basolateral membrane. This potential drives the release of Cl$^-$ through Cl$^-$ channels into the interstitial fluid. Excess HCO$_3^-$ generated by carbonic anhydrases is released across the basolateral membrane via a basolateral Cl$^-$/HCO$_3^-$ exchanger.

Fluid absorption at E14.5 was found to be insensitive to bafilomycin (*Figure 1*). This finding is consistent with minimal v-ATPase activity at E14.5, which is evident from the observation that the luminal pH at E14.5 is only 0.1 pH-units more acidic than the interstitial fluid (*Kim and Wangemann, 2011*). The activity of v-ATPase appears to increase with development, leading to a luminal pH at E17.5 that is 0.55 pH-units more acidic than the interstitial fluid. Activation of v-ATPase would enhance the rate of SLC26A4-mediated Cl$^-$ absorption across the apical membrane through luminal acidification and a steepened HCO$_3^-$ gradient. It is conceivable that v-ATPase, which is expressed at E14.5 (*Li et al., 2013b*), is located mainly in sub-apical membrane vesicles that are subsequently trafficked to the apical membrane later in development. This mechanism has been reported for other epithelia expressing v-ATPase including α-intercalated cells of the renal collecting duct and clear

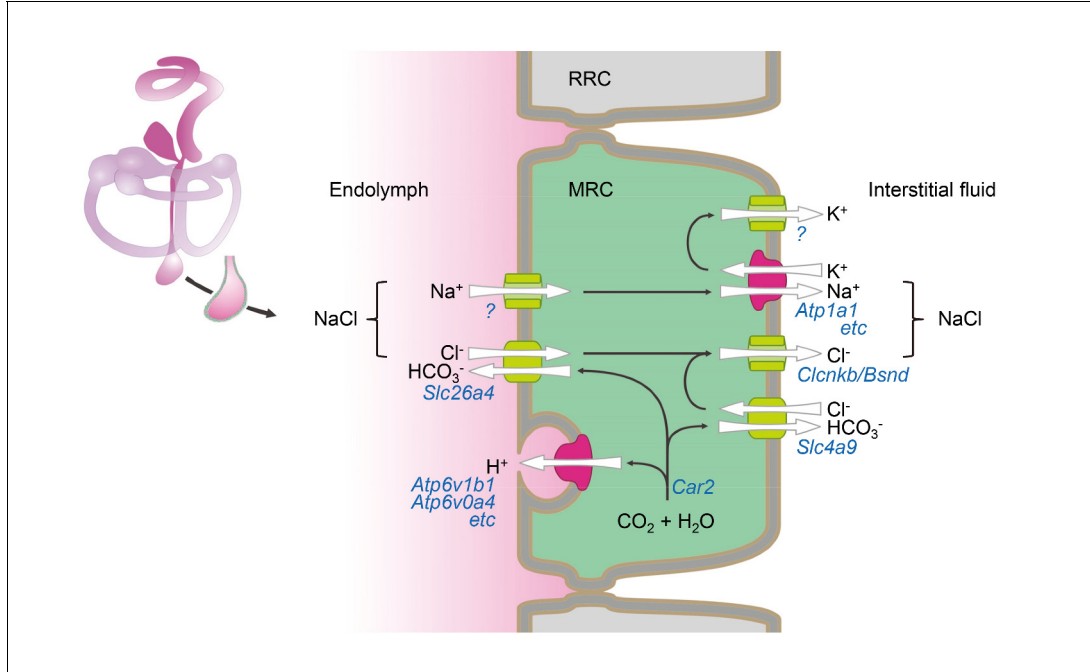

**Figure 9.** Hypothetical model of NaCl absorption by mitochondria-rich cells of the endolymphatic sac. Mitochondria-rich cells (MRC) are interspersed among ribosomal-rich cells (RRC). Ion pumps including $Na^+/K^+$ ATPase and v-ATPase are colored (*red*) different from channels and transporters (*green*). The direction of ion transport is indicated (*arrows*). Ion pumps, channels and transporters are annotated with their corresponding genes. Immunolocalization data reported by other published studies or our current study are summarized in *Figure 9—source data 1*.

DOI: https://doi.org/10.7554/eLife.26851.025

The following source data is available for figure 9:

**Source data 1.** Summary of immunolocalization of proteins in mitochondria-rich cells.

DOI: https://doi.org/10.7554/eLife.26851.026

cells of the epididymis that, like MRCs, belong to the FORE family (*Liu et al., 2012*; *Păunescu et al., 2010*; *Belleannée et al., 2010*). The v-ATPase-dependent increase in fluid absorption appears to be required for normal inner ear development, consistent with the association of EVA with mutations of *ATP6V1B1* and *ATP6V0A4* and the observation of EVA in *Atp6v0a4*$^{\Delta/\Delta}$ mice (*Lorente-Cánovas et al., 2013*; *Hennings et al., 2012*; *Stover et al., 2002*).

Fluid absorption depends on NaCl absorption and thus requires pathways for $Na^+$ and $Cl^-$ absorption. SLC26A4 appears to be required for apical absorption of $Cl^-$. The molecule mediating $Na^+$ absorption, however, remains unclear. $Na^+$ absorption could be transepithelial, paracellular, or both. The driving force for moving $Na^+$ through a paracellular pathway consists mainly of the transepithelial potential. A lumen-positive transepithelial potential can drive paracellular $Na^+$ absorption, whereas a lumen-negative potential would drive $Na^+$ in the opposite direction. Observations of lumen-negative transepithelial potentials in endolymphatic sacs of embryonic *Slc26a4*$^{\Delta/+}$ and *Slc26a4*$^{\Delta/\Delta}$ mice argue against the hypothesis of a paracellular route of $Na^+$ absorption (*Li et al., 2013a*; *Kim and Wangemann, 2011*). In adulthood, however, the situation may be different since small lumen-positive transepithelial potentials have been recorded in adult guinea pigs (*Couloigner et al., 1998*; *Mori and Wu, 1996*).

Transcellular $Na^+$ absorption occurs in a variety of epithelia. Key features of some of the best understood mechanisms were used to guide the present study. Transcellular NaCl transport in the proximal renal tubule occurs via parallel $Na^+/H^+$ and $Cl^-/HCO_3^-$ exchangers with NHE3 being the molecule mediating $Na^+/H^+$ exchange (*Aronson, 2006*). The hypothesis that NHE3 is involved in fluid absorption in the endolymphatic sac can be dismissed by our observation that fluid transport was insensitive to S3226 (*Figure 1*).

Many epithelia, including principal cells of the renal collecting duct, employ a $Na^+$-permeable channel in the apical membrane and a combination of $Na^+/K^+$ ATPase and $K^+$ channels in the

basolateral membrane. A consequence of the localization of these ion channels is a lumen-negative transepithelial potential. The observations of lumen-negative transepithelial potentials in embryonic endolymphatic sacs provide support for this model (*Li et al., 2013a*; *Kim and Wangemann, 2011*). The $Na^+$-permeable channel could be a $Na^+$-selective channel or a non-selective cation channel. One candidate for a $Na^+$-selective channel is ENaC. ENaC is a benzamil-sensitive epithelial $Na^+$-selective channel that requires three subunits encoded by *Scnn1a*, *Scnn1b* and *Scnn1g*, proper intracellular trafficking, and protein modifications for channel activity. The observation that fluid absorption was insensitive to benzamil argues against a role of ENaC in fluid absorption at E14.5 (*Figure 1*). This observation is consistent with our detection of expression of only two (*Scnn1a* and *Scnn1b*) of the three genes encoding the subunits required for ENaC activity. However, ENaC $Na^+$ channels have been shown to contribute to $Na^+$ absorption in the adult human endolymphatic sac (*Kim et al., 2009*).

The $Na^+$-permeable channel could also be a non-selective cation channel. Many non-selective cation channels are sensitive to gadolinium. Our observation that fluid absorption was sensitive to gadolinium provides support for the hypothesis that the $Na^+$ absorption pathway consists of a non-selective cation channel (*Figure 1*). The molecular identity of this channel, however, remains unclear. It is conceivable that the two ENaC subunits, encoded by *Scnn1a* and *Scnn1b*, expressed in the developing endolymphatic sac form a non-selective cation cannel that is gadolinium-sensitive but benzamil-insensitive (*Fyfe and Canessa, 1998*; *Awayda et al., 2004*). Studies of adult endolymphatic sac tissues have provided evidence for $Na^+$ permeable channels and $Na^+$ influx mechanisms into MRCs (*Mori and Wu, 1996*; *Kim et al., 2009*; *Miyashita et al., 2007*), as well as the presence of purinergically controlled non-selective cation channels (*Wu and Mori, 1999*; *Miyashita et al., 2001*). The mechanism of $Na^+$ absorption in the embryonic endolymphatic sac thus remains to be determined.

The $Cl^-$ channel subunits CLCNKB and BSND may not be essential elements for NaCl absorption. EVA has not been associated with hearing loss in Bartter syndrome and endolymphatic enlargement was not observed in $Bsnd^{\Delta/\Delta}$ mice (*Rickheit et al., 2008*). The loss of hearing is instead associated with a loss of function of CLCNKB and BSND in the stria vascularis. There may be another molecular pathway for $Cl^-$ absorption that works independently or redundantly with CLCNK and BSND in the basolateral membrane of MRCs.

The observations that the mechanism of NaCl absorption in the endolymphatic sac has some similarity to that in intercalated cells of the renal collecting duct is not surprising since both cell populations are members of the FORE family (*Hulander et al., 2003*; *Vidarsson et al., 2009*). The renal collecting duct epithelia consist of at least three cell types, α intercalated cells, β intercalated cells, and principal cells (*Kleyman et al., 2013*). The α intercalated cells are mainly responsible for urinary acidification through v-ATPase on the apical membrane and anion exchanger 1 SLC4A1 (AE1) on the basolateral membrane. The β intercalated cells are engaged in $Na^+$ absorption via SLC26A4 and $Na^+$-dependent $Cl^-$/$HCO_3^-$ exchanger SLC4A8 (NDCBE) in the apical membrane and v-ATPase and anion exchanger 4 SLC4A9 (AE4) in the basolateral membrane (*Leviel et al., 2010*; *Gueutin et al., 2013*). CLCNKB (Clc-Kb) and BSND (Barttin) expression are also detected on the basolateral membrane of both α and β intercalated cells (*Estévez et al., 2001*).

Our data highlight several differences between MRCs and intercalated cells: (1) we could not detect any subtype classification among MRCs; (2) we and others cannot detect expression of v-ATPase in the basolateral membrane of MRCs (*Dou et al., 2004*; (3) expression of *Slc4a8* which, in conjunction with *Slc26a4*, plays a crucial role in NaCl absorption in β-intercalated cells, is expressed at low levels in a cell-nonspecific pattern in the endolymphatic sac; and (4) *Scnn1a*, encoding the α subunit of the epithelial sodium channel (ENaC), a marker of principal cells, is exclusively expressed in MRCs. It appears that the molecular components for ion transport in the endolymphatic sac are consolidated into only one cell type, MRCs, and that the endolymphatic sac is a less complex system than the renal collecting duct. This is not surprising given the requirement of the kidney to regulate a much higher dynamic range of fluid and ion homeostasis.

Fluid absorption requires not only transport of ions but also the permeation of water. In the renal collecting duct, water reabsorption is controlled by expression of AQP2, AQP3 and AQP4 in principal cells (*Pearce et al., 2015*). Our data demonstrate that mRNA expression levels of aquaporin family members are quite low in the endolymphatic sac. In our single-cell RNA-seq analysis, only two to five cells of a total of 213 cells expressed *Aqp1*, *Aqp3*, *Aqp4*, *Aqp6*, *Aqp7*, or *Aqp11*, and the

expression levels were low. Our microarray study analysis detected expression of *Aqp1*, *Aqp3*, *Aqp8* and *Aqp11* in *Slc26a4*$^{\Delta/+}$ mice at ages E14.5, E16.5 and E17.5. However, none of these genes showed significant differences in their mRNA levels among time points or between *Slc26a4* genotypes, and they all exhibited much lower expression levels in comparison to those of ion transporter and channel genes (*Figure 8A*). There are at least three possible explanations for our observations. First, aquaporin proteins may be stable with only low copy numbers of mRNA. However, a proteome-wide study (*Sandoval et al., 2013*) of proteins expressed in vasopressin-sensitive mouse collecting duct cell line (mpkCCD cells) indicated that the half-life of AQP2 is shorter than the median half-life of 3691 proteins (9.25 hr vs. 31.2 hr). Half-lives of other aquaporin proteins were not reported in that study. Second, incubation of the endolymphatic sacs in buffer during dissection and cell dissociation may lead to a selective decrease in aquaporin mRNA abundance. Third, the endolymphatic sac epithelium may not require the enhanced permeability of water conferred by aquaporin expression. Considering the relatively small volume of the developing inner ear and the absorptive surface area of the endolymphatic sac, the flux of water traversing the epithelium may be much smaller in the endolymphatic sac than in the kidney. The volume requirement for fluid absorption in the developing sac must also depend on the rate of fluid secretion into the developing endolymph, which is unknown (*Kim and Wangemann, 2010*). Therefore, it is possible that diffusion of water via transcellular pathways, paracellular routes, or both, in combination with permeation through a small number of aquaporin channels may be sufficient for normal development.

Our results are consistent with previous observations that SLC26A4 is equally distributed across the entire endolymphatic sac (*Kim and Wangemann, 2011*). It is interesting that we observed a subset of RRCs with low expression of *Slc26a4* (*Slc26a4*-positive RRCs). Principal component analysis or hierarchical clustering did not separately cluster the *Slc26a4*-positive RRCs from other RRCs, presumably because of the absence of genes whose expression strongly correlated with *Slc26a4* levels in RRCs. Inspection of the RNA-seq read alignments indicates there are no unique or different isoforms in the *Slc26a4*-positive RRCs. Remaining challenges include determining whether the presence of *Slc26a4* mRNA and SLC26A4 protein in RRCs is functionally important, a result of more promiscuous expression, or a reflection of a transitional differentiation state between RRCs and MRCs.

Interpretation of single-cell RNA-seq data has caveats. First, the enzymatic dissociation of tissue, the length of time from dissection to cell lysis, or both may cause transcriptional changes. Second, the Fluidigm C1 chip that we used is designed to capture cells ranging from 10 to 17 μm in diameter. Therefore, we may have missed larger or smaller cells. Third, single-cell RNA-seq might not detect functionally important genes expressed at levels below the limit of detection. Fourth, our strategy to detect differentially expressed genes may have missed functionally important genes expressed at similar levels in both cell types. Our microarray approach compensates for these shortcomings of single-cell RNA-seq analysis. In order to look for functionally important genes in the endolymphatic sac, we used profiles of averages of mixed cell populations, instead of comparing cell-type specific genes among cell types as we did in single-cell RNA-seq. Thus, the microarray approach, in combination with qPCR and immunohistochemistry analyses of expression of a subset of genes, supports the validity of our RNA-seq results and interpretations.

In conclusion, our data provide a comprehensive model for the normal function and differentiation of the developing endolymphatic sac. The numerous new details of our model should be further tested experimentally. The results will enhance our understanding of the structure of the endolymphatic sac and reveal potential opportunities to modulate this pathway in order to prevent or treat loss of hearing or balance associated with EVA and other diseases associated with endolymph homeostasis.

## Materials and methods

### Animals

A colony of C57BL/6J mice (RRID:IMSR_JAX:000664) was established at the National Institute on Deafness and Other Communication Disorders (NIDCD) with animals obtained from the Jackson Laboratory (Bar Harbor, Maine). In addition, a colony of *Slc26a4*$^{+/+}$, *Slc26a4*$^{\Delta/+}$ and *Slc26a4*$^{\Delta/\Delta}$ mice (RRID:IMSR_JAX:018424) (*Everett et al., 2001*) on the 129S6 background was maintained at Kansas State University-College of Veterinary Medicine. The colony at Kansas State University was

established with breeders kindly provided by Dr. Susan Wall (Emory University, Atlanta,GA). Genetic drift of the colony was limited by periodic back-crossing to the parental strain 129S6/SvEvTac (RRID: IMSR_TAC:129sve, Taconic, Germantown, NY). Genotyping of $Slc26a4^{\Delta}$ and $Slc26a4^{+}$ alleles was performed by Transnetyx (Memphis, TN).

$Slc26a4^{\Delta/+}$ and $Slc26a4^{\Delta/\Delta}$ mice ranging in age from embryonic day 13.5 (E13.5) to E17.5 were used in the present study. Time-pregnant dams were deeply anesthetized with 4% tri-bromo-ethanol (0.014 ml/g body weight, i.p.) before harvesting embryos by sterile laparotomy. Dams and embryos were sacrificed by decapitation. The day when a vaginal plug was detected was considered to be E0.5.

## In vitro measurements of fluid absorption

Endolymphatic sacs were isolated by micro-dissection of E14.5 $Slc26a4^{\Delta/+}$ and $Slc26a4^{\Delta/\Delta}$ mice of either sex using sharpened forceps and a stereo microscope (MZ16, Leica, Wetzlar, Germany). The lumen was filled with solution either via the cochlea or the endolymphatic duct using sharpened glass pipettes (20 μm tip OD) and a manual injection system (Cell Tram vario, Eppendorf, Hamburg, Germany). The fluorescent dye seminaphtarhodafluor (SNARF-1) appears dark red, which aided the filling procedure. Endolymphatic sacs were sealed manually using fine glass needles, transferred individually to glass-bottom dishes (P35G-1.5–10 C, MatTek, Ashland, MA) holding ~150 μl of culture medium (DMEM, 12800–058, GIBCO, Gaithersburg, MD) supplemented with 2.3 g/l $NaHCO_3^-$ and 1% Pen-Strep (A-5955, Sigma, St Louis, MO) and incubated at 37°C in 5% $CO_2$ (HeraCell 240, Heraeus, Hanau, Germany).

Three groups of experiments were performed. In the first group of experiments, we evaluated fluid absorption in E14.5 $Slc26a4^{\Delta/+}$ and $Slc26a4^{\Delta/\Delta}$ mice with and without abluminal administration of ouabain (1 mM, O3125, Sigma). Endolymphatic sacs were filled with a solution containing (in mM) 137 NaCl, 7 KCl, 12.5 $NaHCO_3^-$, 0.5 $MgCl_2$, 0.35 $CaCl_2$, 2.5 glucose and 2.5 seminaphtarhodafluor (SNARF-1, C1270, Invitrogen, Carlsbad, CA).

In the second group of experiments, we evaluated fluid absorption in E14.5 $Slc26a4^{\Delta/+}$ endolymphatic sacs in the absence and presence of luminal benzamil (10 μM or 100 μM, B2417, Sigma), bafilomycin A1 (10 nM or 10 μM, B1793, Sigma) or S3226 (20 μM, a kind gift of Sanofi-Aventis, Germany). Stocks of benzamil, bafilomycin A1 and S3226 were prepared in dimethyl sulfoxide (DMSO) before dilution in luminal solution containing (in mM) 137 NaCl, 7 KCl, 12.5 $NaHCO_3^-$, 0.5 $MgCl_2$, 0.35 $CaCl_2$, 2.5 glucose, and 2.5 SNARF-1. The final concentration of DMSO was 0.1% in control experiments, 0.05% in benzamil experiments, and 0.1% in bafilomycin A and S3226 experiments.

In the third group of experiments, we evaluated fluid absorption in E14.5 $Slc26a4^{\Delta/+}$ mice in the absence and presence of luminal gadolinium (2 mM, G7532, Sigma). Endolymphatic sacs were filled with a solution containing (in mM) 135 NaCl, 4 KCl, 25 $NaHCO_3^-$, 1 $MgCl_2$, 0.7 $CaCl_2$, 5 glucose, and 2.5 SNARF-1.

For all three groups of experiments, endolymphatic sacs were imaged semi-hourly and the luminal volume was determined by 3D confocal microscopy. Imaging consisted of scanning microscopy and superimposition of transmission images recorded with red (633 nm), green (561 nm) and blue (405 or 458 nm) lasers (20x/0.8 objective; custom hardware configuration of LSM 880, Carl Zeiss, Jena, Germany). For 3D microscopy, SNARF-1 was excited at 488 nm and fluorescence intensity was recorded between 604 and 648 nm via a GaAsP detector (LSM 880, Carl Zeiss). Dimensions of individual voxels were $3 \times 3 \times 3$ μm resulting in a voxel volume of 27 $μm^3$ or $2.7 \times 10^{-5}$ nl. Laser exposure times (3.88 μs per voxel) were chosen to be short to speed data acquisition and minimize photodamage. Post-acquisition 3D images were smoothed (xyz: 5 in Zen Black, Carl Zeiss). Luminal voxels were isolated by setting a threshold and counted using a custom-written macro (Zen Black, Carl Zeiss). Voxel counts were converted to luminal fluid volumes (nl). The rate of absorption was determined from the linear portion of a graph of luminal volume plotted against time. Rates of absorption (nl x $hr^{-1}$) were normalized to the estimated luminal surface area. The luminal surface area of the endolymphatic epithelium was estimated from the first 3D image taken during an experiment. For geometric simplification, the luminal volume was modeled as two spherical caps that enclose a frustum of a cone (*Figure 1—figure supplement 1A*). Surfaces of the spherical caps and of the frustum were estimated from simple linear dimensions (*Figure 1—figure supplement 1B*).

Normalized data were presented as nl x hr$^{-1}$ x mm$^{-2}$. Grubbs' outlier test was performed and one outlier was removed from the dataset obtained from endolymphatic sacs of *Slc26a4*$^{\Delta/\Delta}$ mice (*Figure 1—source data 1*). Normal distribution of the data was evaluated by the Shapiro-Wilk test. Power and significance were evaluated by one-way ANOVA followed by an all pairwise multiple comparison procedure (Bonferroni t-test). Shapiro-Wilk, Brown-Forsythe, and one-way ANOVA were conducted in SigmaStat 4.0 (RRID:SCR_010285, Systat Software, San Jose, CA). Grubbs' outlier test and Cohen's D were calculated in Excel 2010 (Microsoft).

## Cell isolation

Single-cell suspensions of endolymphatic sac epithelia were prepared from E12.5, E16.5, P5 or P30 C57BL/6J mice of both sexes for RNA-seq or qPCR analysis. Inner ears were removed from four to five mice and placed in ice-cold DMEM/F12 (GIBCO). The endolymphatic sac was harvested together with surrounding connective and osseous tissues. The tissues were pooled and incubated in 1 mg/ml collagenase (LS004194, Worthington, Lakewood, NJ) and 1 mg/ml dispase (LS02100, Worthington) in DMEM/F12 at 37°C for 10 min. The tissues were then transferred into 10% fetal bovine serum (FBS) (GIBCO) in DMEM/F12 where the epithelia were isolated from surrounding tissues. The epithelia were transferred to 50 µl of 0.125% trypsin/EDTA (GIBCO) in phosphate-buffered saline (PBS) in a 1.5-mL tube and incubated for 15 min at 37°C. Then 50 µL of 10% FBS in DMEM/F12 was added to the tube to stop digestion. The epithelia were gently triturated with a 200-µl micropipette tip to complete the dissociation. Cell concentration was measured using Luna-FL automated cell counter (Logos Biosystems, Annandale, VA) and adjusted to 250,000 to 300,000 cells/ml by adding 5% FBS in DMEM/F12. To assess cell viability after cell capturing, the cells were labeled with 1:1000 of calcein-AM or ethidium homodimer-1 (EthD-1) (LIVE/DEAD cell viability assay, Thermo Fisher Scientific, Waltham, MA) and placed on ice until capture.

## Microfluidic capture of single cells

Cell capture, lysis, reverse transcription (RT), and PCR amplification of cDNA were performed in accordance with the Fluidigm protocol (PN 100–5950 B1) and methods described previously (*Burns et al., 2015*). Briefly, 12 µl of the single-cell suspension was loaded onto a medium-sized (10–17 µm) C1 Single-Cell Auto Prep IFC for mRNA Seq (Fluidigm, South San Francisco, CA), where single cells were automatically washed and captured. After capturing, the chip was removed to acquire Z-stack fluorescence and brightfield images of each of the 96 capture sites using a 10/0.4 numerical aperture objective on an inverted Axio Observer Z1 microscope (Carl Zeiss). Lysis, RT, and PCR mixes were prepared from the SMARTer Ultra Low RNA kit (Clontech, Palo Alto, CA) and loaded into the chip. We added Ambion External RNA controls Consortium (ERCC) RNA spike-in control mix 1 (Themo Fisher Scientific) to the cell lysis solution at a concentration of 1:20,000 for time points E16.5 and P5. The chip was then returned to the C1 instrument where lysis, RT, and PCR were performed. cDNA was manually collected from the output channel of each capture site and stored in a 96-well plate at 20°C until library preparation.

## Preparation of pooled cell populations

For time points E16.5 and P5, we prepared cDNA of pooled cell populations using the leftover single-cell suspension in accordance with the Fluidigm protocol. Briefly, cells were diluted to a final concentration of 150,000 cells/ml and stored on ice. Twenty microliters of cells were lysed in the C1 instrument, and RNA was eluted into 20 µl of water using RNeasy Plus Micro Kit (Qiagen, Hilden, Germany). Then 1 µl of RNA corresponding to approximately 150 cells was used for cDNA preparation with the same mixes that were used for single cells. Two replicates of pooled cell cDNA were generated for each single-cell capturing experiment.

## Identification of cDNA from single cells

Cell number of each capture site was manually counted from the Z-stack images. Average intensity of fluorescence of calcein-AM (for live cells) and EthD-1 (for dead cells) in single cells was acquired using a custom macro program of ImageJ and normalized to a range of zero to 1000. We used arbitrary thresholds of 10 to designate a stain result as positive. We considered cells as 'live' if they were calcein-AM$^{+}$/EthD-1$^{-}$, or if they were calcein-AM$^{+}$/EthD-1$^{+}$ but EthD-1 staining was limited to a small

area. The concentration of cDNA obtained from each capture site was measured in duplicate using a PicoGreen assay (Thermo Fisher Scientific) and a microplate reader (DTX 880 multimode detector, Beckman Coulter, Miami, FL). For sequencing, we selected single, live cells with a cDNA concentration of >0.75 ng/ml.

During the course of the study, we received a report from Fluidigm that the medium-size microfluidic chips showed an unexpectedly high rate of cell doublets. We re-evaluated z-stack images of single cells for which we had already generated sequence data and excluded all suspected doublets or samples whose cell counts could not be established due to insufficient focusing (*Figure 2—source data 2*).

## RNA-seq library construction, sequencing and mapping

cDNA was diluted to a final concentration of 0.1 to 0.3 ng/ml if the initial concentration was >0.3 ng/ml. Libraries were constructed in 96-well plates using the Nextera XT DNA Sample Preparation kit (Illumina, San Diego, CA) according to the Fluidigm protocol. Barcoded libraries were pooled and purified using Agencourt AmPure XP beads (Beckman Coulter). Libraries were quantified on an Agilent 2100 Bioanalyzer using a High Sensitivity DNA analysis kit (Agilent Technologies, Santa Clara, CA).

Libraries were sequenced on an Illumina HiSeq instrument, generating 125 × 125 paired-end (PE) reads. The PE reads were trimmed with Trimmomatic (*Bolger et al., 2014*) to remove adapter and low-quality sequences, and mapped with the STAR v2.4 aligner (RRID:SCR_005622) (*Dobin et al., 2013*) to the mouse genome (mm10; NCBIGRCm38), generating output in transcriptome coordinates (i.e. 'quantMode TranscriptomeSAM' option). Transcript abundance was calculated with RSEM (RRID:SCR_013027) (*Li and Dewey, 2011*). RNA-Seq data have been deposited in the NCBI GEO database under accession code GSE87293.

## Data pre-processing

Data transformation, computational outlier identification, and cross-sample normalization were performed as described previously (*Burns et al., 2015*). Briefly, relative transcript abundances (transcripts per million, TPM) were converted to abundances at the gene level by summing the estimated transcript abundances for each gene. Then, we performed outlier identification analysis for cells at each age using SINGuLAR Analysis Toolset 3.0 (RRID:SCR_15685, Fluidigm). After excluding outliers (*Figure 2—source data 2*), cross-sample normalization was conducted using the DESeq analysis packages (RRID:SCR_000154), in which normalized TPM (nTPM) was calculated for each gene. Levels of nTPM <1 were set to zero and nTPM levels > 1 were transformed to $\log_2$ (nTPM), except in the single-cell trajectory analysis described below.

## Clustering of single cells

Custom R scripts (*Burns et al., 2015*) were used for principal component analysis and hierarchical clustering analysis. All genes expressed (nTPM >1) in more than two cells with a cell-to-cell coefficient of variation >0.5 were used for principal component analysis (PCA). For the subsequent hierarchical clustering analysis, we selected a limited number of genes that were identified by PCA to account for the majority of the variance between the cells. We empirically determined the number of genes and which PCs to be used in the hierarchical clustering, based on results of PCA.

## Differential expression analysis

Differential expression testing was performed using custom R scripts (*Burns et al., 2015*) with modifications of the Monocle and cummeRbund packages (*Goff et al., 2013*). The criteria for significant differential expression were: (1) false discovery rate (FDR) < 0.05 after the Benjamini-Hochberg correction; (2) detection in more than 50% of cells in the relevant group; and (3) specificity score >0.5 or 0.65 for comparisons between two groups or four groups, respectively. Specificity scores range from 0 to 1, with 0 being least specific and one being completely specific.

## Gene-ontology enrichment analysis

Gene-ontology enrichment analysis was performed using a tool from the PANTHER Classification System (RRID:SCR_004869) in the Gene Ontology Consortium website (http://geneontology.org)

(*Gene Ontology Consortium, 2015*). The GO Ontology database (released 2016-05-20) was used for the analysis. The fold-enrichment and p-value of a submitted gene-set were calculated by comparison with the distribution of annotation in the *Mus musculus* reference gene-set (22,320 genes) using Fisher's exact test.

## Single-cell trajectory analysis

A single-cell trajectory analysis was performed with Monocle 2. Normalized TPM data of 167 cells at E12.5, E16.5, and P5 were used for the analysis. During the first step, a set of genes were selected whose expression reflected a cell's progress through a biological process and did not reflect batch effects or noise (*Trapnell et al., 2014*). To achieve an unsupervised reconstruction of single-cell trajectories, we tested for differential expression among E12.5, E16.5 and P5 cells (n = 43, 25, and 80), excluding proliferating cells that were identified by the prior analysis, and thus selected 3258 genes with FDR < 0.01 for the subsequent steps. The expression data for the selected genes was projected into two-dimensional space using a DDRTree technique in which the sequence of gene expression is reflected as a 'branched' trajectory along pseudo-time with each cell displayed along the trajectory.

## Single-cell qPCR

For single-cell qPCR, we captured single cells of P5 endolymphatic sacs using medium-sized microfluidic chips (C1 Single-Cell Auto Prep IFC for PreAmp) as outlined in the Fluidigm protocol (PN 100–6117 G1). Mixes for lysis, RT, and specific target amplification were prepared from the Single Cell-to-Ct qRT-PCR kit (Ambion, Austin, TX) and pre-designed TaqMan gene expression assays (Thermo Fisher Sientific). In addition to canonical cell markers, putative MRC and RRC markers at P5 (*Figure 2—source data 3*) were selected for analysis in an arbitrary manner.

After 18 cycles of preamplification, expression levels were measured by qPCR on a 96.96 Dynamic Array IFC using the Fluidigm BioMark HD system. cDNA from single cells was selected for qPCR in the same way as it was selected for RNA-seq. The threshold of cycles ($C_t$) values were calculated with Fluidigm Real-time PCR analysis software (RRID:SCR_15686) with the following settings: quality threshold of 0.65; a linear (derivative) baseline correction; and auto (detectors) method. We defined gene expression levels as $\log_2$ expression = LOD – $C_t$, in which we set $C_t$ = 28 as LOD. We used the $\log_2$ expression dataset for hierarchical clustering.

## Gene array analysis

Endolymphatic sacs were obtained by manual micro-dissection from $Slc26a4^{\Delta/+}$ and $Slc26a4^{\Delta/\Delta}$ mice at ages E13.5, E14.5, E16.5 and E17.5. Microdissection was carried out in a cooled (4°C) physiologic salt solution containing (in mM): 150 NaCl, 1.6 $KH_2PO_4$, 0.4 $K_2HPO_4$, 0.7 $CaCl_2$, 1.0 $MgCl_2$ and 5 glucose, pH 7.4. Endolymphatic sacs were isolated by manual micro-dissection using sharpened forceps and a high-power stereo microscope (MZ16, Leica). Endolymphatic sacs were dissected free of connective tissue and removed from endolymphatic ducts. Total RNA was extracted (RNeasy Plus Micro, Qiagen) and stabilized with 0.33 mM Na-citrate (RNA storage solution, Applied Biosystems/ Ambion). The RNA integrity number (RIN) was used to rate the quality of isolated RNA on a scale from 1 (most degraded) to 10 (most intact). Quantity and quality by RIN was determined by RT-PCR of 18S rRNA and microfluidic electrophoresis (2100 Bioanalyzer, RNA 6000 Pico, Agilent), respectively (*Dou et al., 2003*). RNA samples forwarded to target generation were prepared in volumes of 5 µl with a final Na-citrate concentration ranging from 0.2 to 2.1 mM. Quantity and quality of total RNA analyzed on gene array chips was tabulated (*Figure 8—source data 1*).

Total RNA was reverse transcribed into cDNA, amplified, fragmented and labeled following manufacturer's procedures (Ovation Pico WTA System V2 and Encore Biotin, NuGen, San Carlos, CA). Following fragmentation and labeling, ~6 µg of cDNA was hybridized (GeneChip Hybridization Oven 640, Affymetrix, Santa Clara, CA) to gene arrays (Mouse Genome 430 2.0 GeneChip, Affymetrix), washed and stained (GeneChip Fluidics Station 450, Affymetrix) according to the manufacturer's procedures. Data were processed using GCOS software (RRID:SCR_003408, Affymetrix) using default parameters. The scaling factor was set up to 500. No further normalization was performed at this stage of the analysis. Standard quality controls, including scaling factors, average intensities, present calls, background intensities, noise, raw Q values, *Actb* 3'/5' and *Gapdh* 3'/5' ratios were all within acceptable limits (*Figure 8—source data 1*).

We evaluated the variability between three biological replicates for the same age and genotype. Scatterplot matrices of signal intensities of all probes showed a high correlation between any pair of the three replicates (Pearson correlation = 0.92–0.95) except for replicates of $Slc26a4^{\Delta/+}$ mice at E13.5 (*Figure 8—figure supplement 1A*). Further, we evaluated the purity of the endolymphatic sac preparations by inspecting the expression of markers for epithelial (*Epcam* and *Cdh1*, encoding E-cadherin), endothelial (*Cd34*) and mesenchymal cells (*Cd44*) (*Figure 8—figure supplement 1B*). One replicate of $Slc26a4^{\Delta/+}$ mice at E13.5 (Chip 5) showed lower signal intensities of epithelial cell markers and higher intensities of endothelial and mesenchymal cell, indicating the presence of contaminating non-epithelial tissues; therefore, data from this chip was excluded from further analysis. All data, however, are available via the GEO Database (Accession# GSE90092).

Two strategies of analysis were used. One approach started with cell models of ion transport to identify important genes within the expression data set. The other was a regression-based approach that started with the whole expression data set, filtered the data, and used hierarchical clustering to group genes. For the model-based approach, the data were loaded into a Microsoft Excel spreadsheet, annotated (ftp://ftp.informatics.jax.org/pub/reports/Affy_430_2.0_mgi.rpt downloaded May 2, 2015) and analyzed with custom-written macros (*Jabba et al., 2006*). Probes were retained when all three replicates of $Slc26a4^{\Delta/+}$ and $Slc26a4^{\Delta/\Delta}$ mice at ages E14.5, E16.5 and E17.5 called 'present' and when the sum of the average intensity values was $\geq$600, which is a factor of approximately two above background intensity. Cell models for NaCl absorption were constructed with features of known models of $Na^+$ and $Cl^-$ absorption in epithelia such as the renal thick ascending limb, the renal connecting tubule, and the renal collecting duct. Models were rejected when critical genes were not expressed, when expression levels of critical genes did not follow the expected positive correlation with the onset of fluid absorption, or when models were incompatible with clinical observations or functional studies in mouse models.

For the regression-based approach, gene array data were normalized using the quantile normalization function in the limma package available from Bioconductor (*Ritchie et al., 2015*). Probes which were called 'present' in all three replicates of $Slc26a4^{\Delta/+}$ mice at ages E14.5, E16.5 and E17.5 were retained (23,699 probes). Signal intensities were transformed to $\log_2$ scale. To investigate the differences in temporal gene expression profiles between genotypes, a regression-based approach was performed using maSigPro (RRID:SCR_001349), a package for time-course microarray analysis available from Bioconductor (*Conesa et al., 2006*). We defined a regression model where the dependent variable was the signal intensity of each probe and the independent variables were time (linear and quadratic) and genotype (binary variable). Both groups were subjected to hierarchical clustering (using maSigPro). Probe IDs were then converted to gene names with DAVID (RRID:SCR_001881) (*Huang et al., 2009a*; *Huang et al., 2009b*). When any one of multiple probes mapping to the same gene were statistically significant, we considered the gene as significant. When multiple probes in the same gene were assigned to different clusters, we classified the gene to the smallest number cluster.

For comparison of gene array data with single-cell RNA-seq data, we used a dataset of two or three replicates from $Slc26a4^{\Delta/+}$ endolymphatic sacs at ages E13.5 or E16.5, and a dataset of 44 or 41 single cells from wild-type endolymphatic sacs at ages E12.5 or E16.5 respectively. In gene array, data for probes was retained when all three replicates of $Slc26a4^{\Delta/+}$ at E16.5 were called 'present' and when the sum of the average intensity values was $\geq$150. In single-cell RNA-seq analyses, data for genes was retained when the average expression levels for 41 single cells at E16.5 were >1 nTPM.

## RT-qPCR

Endolymphatic sacs were obtained from C57BL/6J mice at E12.5, E14.5, E16.5, E18.5, P0, P2, P5, P10, P15, P30, or P90. Both sacs were harvested with adjacent connective and osseous tissues from each animal, and the epithelia were isolated as described in the method section 'cell isolation'. Tissues were lysed and RNA was extracted using the Arcturus PicoPure RNA isolation kit (Thermo Fisher Scientific). Quantity of total RNA was measured with the Qubit RNA HS assay kit (Thermo Fisher Scientific) and 10 ng of total RNA was reverse-transcribed to cDNA with the SuperScript VILO cDNA Synthesis Kit (Thermo Fisher Scientific).

RT-qPCR was performed using the Fluidigm BioMark HD system as outlined in the manufacturer's protocol. Briefly, cDNA was pre-amplified for 14 cycles with Preamp Master Mix (Fluidigm) and pre-

designed TaqMan gene expression assays (Thermo Fisher Scientific). Real-time PCR analyses were performed on a 96.96 Dynamic Array IFC under default PCR conditions. The threshold-of-cycles ($C_t$) values were calculated with Fluidigm Real-time PCR analysis software with a quality threshold of 0.65, a linear (derivative) baseline correction, and the auto (detectors) method. For each assay, standard curves were generated using serial five-fold dilutions of mixed cDNA from mouse endolymphatic sac at E18.5 and P5 and from adult mouse kidney. The efficiency of amplification with each assay was calculated from the slope of the standard curve: efficiency = $10^{-1/slope}$. The relative change in mRNA level was determined by comparison with the mean value of E16.5 $C_t$, after adjustment to the value for *Actb*: ratio = $[(\text{efficiency}_{\text{target gene}})^{\Delta Ct(\text{E16.5 - Sample})} / (\text{efficiency}_{Actb})^{\Delta Ct(\text{E16.5 - Sample})}]$.

## Immunohistochemistry

For immunohistochemistry of ATP1A1, endolymphatic sacs were isolated by microdissection and fixed with methanol for 1 hr. After fixation, sacs were permeabilized and blocked in phosphate-buffered saline (PBS) with 0.15% Triton X-100 (PBS-TX) and 5% bovine serum albumin (BSA) at room temperature. Sacs were then washed three times with PBS-TX and incubated overnight at 4°C with an Alexa-488-conjugated anti-ATP1A1 mouse monoclonal antibody (1:100, RRID:AB_2713944, ab197496, Abcam, Cambridge, MA) and DAPI (1:1000, D3571, Thermo Fisher) in PBS-TX with 2.5% BSA. Sacs were then washed three times with PBS-TX and mounted between two coverslips with FluorSave (345789, Calbiochem, Billerica, MA) and imaged using a Zeiss LSM 880 confocal microscope. Averaged staining intensities were obtained in regions of interest of the endolymphatic sac or the endolymphatic duct (Zen Blue, Carl Zeiss). Regions of interest averaged 68,215 $\mu m^2$ in the endolymphatic sac and 32,905 $\mu m^2$ in the endolymphatic duct. Average intensity of ATP1A1 staining was normalized to the average intensity of the corresponding DAPI staining and ATP1A1 expression was compared between endolymphatic sacs and ducts by calculating the ratio of normalized ATP1A1 staining.

For immunohistochemistry of SLC26A4, ATP6V1B1/2, FOXI1, BSND, and CFTR, temporal bones were harvested and fixed in fresh 4% paraformaldehyde in PBS for 30 min at room temperature. After fixation, endolymphatic sacs were isolated by microdissection, permeabilized for 30 min in phosphate-buffered saline (PBS) with 0.5% Triton X-100, and blocked for 1 hr with 5% normal donkey serum at room temperature. Samples were then incubated with primary antibodies in PBS with 5% normal donkey serum overnight at 4°C, followed by three rinses in PBS with 0.2% Triton X-100 (PBS-T) and labeling with Alexa Fluor-conjugated secondary antibodies (1:500, Thermo Fisher Scientific) in 5% normal donkey serum for 1 hr at room temperature. Alexa Fluor-conjugated phalloidin (Thermo Fisher Scientific) was included with the secondary antibodies to detect F-actin. Organs were rinsed three times in PBS and mounted in ProLong Gold with 4,6-diamidino-2- phenylindole. Whole-mount specimens were imaged using a Zeiss LSM 780 confocal microscope.

The following antibodies were used: rabbit anti-pendrin (RRID:AB_2713943, PB826) (*Choi et al., 2011*) diluted 1:1000, rabbit anti-V-H$^+$-ATPase B1/B2 (RRID:AB_677577, H-180, Santa Cruz, Dallas, TX) diluted 1:200, goat anti-FOXI1 (RRID:AB_732416, ab20454, Abcam) diluted 1:200, mouse anti-BSND (RRID:AB_2067095, H0007809-B01P, Novus Biologicals, Littleton, CO) diluted 1:50, rabbit anti-AE4 (RRID:AB_1609269, AE41-A, Alpha Diagnostic, San Antonio, TX) diluted 1:200, and rabbit anti-CFTR (RRID:AB_2039804, ACL-006, Alomone, Jerusalem, Israel) diluted 1:100. For double-labeling of SLC26A4 and ATP6V1B1, the Zenon Rabbit IgG Labeling Kit (Z25360, Thermo Fisher Scientific) was used according to the manufacturer's instructions.

## Proliferation assay

Intraperitoneal administrations of 50 µg of EdU (A10044, Thermo Fisher Scientific) per gram body weight were performed for pregnant females or neonatal pups, followed by harvesting of tissues after 2 hr. EdU incorporation was detected with the Click-iT Plus EdU Alexa Fluor 555 imaging kit (C10638, Themo Fisher Scientific) according to manufacturer's instructions.

## Quantification of cells

To estimate cellular density of EdU-positive cells or FOXI1-positive cells, all nuclei stained with DAPI and EdU- or FOXI1- labeled nuclei were counted manually using the Cell Counter plug-in ImageJ. For each specimen, a Z-stack of confocal images of the endolymphatic sac was acquired using a 40x/

1.3-NA objective under the same laser power and gain settings. Adjusting brightness of Z-stacks were held constant across all specimens in each experiment. One slice of the Z-stack was randomly selected for count.

## Study approval

All animal experiments and procedures were performed according to protocols approved by the Animal Care and Use Committees of the National Institute of Neurological Diseases and Stroke/ National Institute on Deafness and Other Communication Disorders (#1264) or by the Animal Care and Use Committee at Kansas State University (#2961, #3245, #3698).

## Acknowledgements

This work was supported by the NIH intramural research program funds Z01-DC000059, Z01-DC000060, Z01-DC000086 and Z01-DC000088. SHK, LC, XL, FZ and PW were supported by NIH grant R01-DC012151. Gene arrays were funded by 1P20RR17686 and processed at the Integrated Genomics Facility at Kansas State University. This work utilized the computational resources of the NIH HPC Biowulf cluster (http://hpc.nih.gov). We thank S Saxena and E Boger for assistance with sequencing, A Akhunova (KSU) for processing gene arrays, T Miesner and M Vilardo (KSU) for animal care, S Raft and I Belyantseva and our NIDCD colleagues for helpful suggestions and advice, K Kitamura for support and encouragement, and T Friedman and D Wu for critical reading of the manuscript.

## Additional information

### Funding

| Funder | Grant reference number | Author |
| --- | --- | --- |
| National Institute on Deafness and Other Communication Disorders | Z01-DC000060 | Andrew J Griffith |
| National Institute on Deafness and Other Communication Disorders | Z01-DC000059 | Matthew W Kelley |
| National Institute on Deafness and Other Communication Disorders | Z01-DC000086 | Robert J Morell |
| National Institute on Deafness and Other Communication Disorders | Z01-DC000088 | Michael Hoa |
| National Institute on Deafness and Other Communication Disorders | R01-DC012151 | Philine Wangemann |
| National Center for Research Resources | P20-RR017686 | Philine Wangemann |

The funders had no role in study design, data collection and interpretation, or the decision to submit the work for publication.

### Author contributions

Keiji Honda, Conceptualization, Data curation, Software, Validation, Investigation, Visualization, Methodology, Writing—original draft, Project administration, Writing—review and editing; Sung Huhn Kim, Laura Constance, Xiangming Li, Fei Zhou, Formal analysis, Investigation, Writing—review and editing; Michael C Kelly, Resources, Data curation, Software, Investigation, Methodology, Writing—review and editing; Joseph C Burns, Resources, Software, Investigation, Methodology, Writing—review and editing; Michael Hoa, Matthew W Kelley, Resources, Supervision, Funding acquisition, Methodology, Writing—review and editing; Philine Wangemann, Conceptualization, Resources, Data curation, Software, Formal analysis, Supervision, Funding acquisition, Validation,

Investigation, Visualization, Methodology, Writing—original draft, Project administration, Writing—review and editing; Robert J Morell, Resources, Data curation, Software, Supervision, Funding acquisition, Writing—original draft, Writing—review and editing; Andrew J Griffith, Conceptualization, Resources, Supervision, Funding acquisition, Writing—original draft, Project administration, Writing—review and editing

### Author ORCIDs
Keiji Honda [iD] https://orcid.org/0000-0002-3411-8539
Michael Hoa [iD] http://orcid.org/0000-0001-7469-2909

### Ethics
Animal experimentation: All animal experiments and procedures were performed according to protocols approved by the Animal Care and Use Committees of the National Institute of Neurological Diseases and Stroke/National Institute on Deafness and Other Communication Disorders (#1264) and by the Animal Care and Use Committee at Kansas State University (#2961, #3245, #3698).

### Decision letter and Author response
Decision letter https://doi.org/10.7554/eLife.26851.036
Author response https://doi.org/10.7554/eLife.26851.037

## Additional files
### Supplementary files
• Source Code 1. R script for analysis and visualization
DOI: https://doi.org/10.7554/eLife.26851.027

• Transparent reporting form
DOI: https://doi.org/10.7554/eLife.26851.028

### Major datasets
The following datasets were generated:

| Author(s) | Year | Dataset title | Dataset URL | Database, license, and accessibility information |
|---|---|---|---|---|
| Honda K, Kelly MC, Burns JC, Hoa M, Morell RJ, Kelley MW, Griffith AJ | 2017 | Molecular architecture underlying fluid absorption by the developing inner ear | http://www.ncbi.nlm.nih.gov/geo/query/acc.cgi?acc=GSE87293 | Publicly available at the NCBI Gene Expression Omnibus (accession no: GSE87293) |
| Wangemann P, Li X, Zhou F | 2017 | Gene expression in the embyonic endolymphatic sac of E13.5, E14.5, E16.5 and E17.5 Slc26a4Δ/+ and Slc26a4Δ/Δ mice | https://www.ncbi.nlm.nih.gov/geo/query/acc.cgi?acc=GSE90092 | Publicly available at the NCBI Gene Expression Omnibus (accession no: GSE90092) |

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
