## [Decision Letter]

Thank you for submitting your manuscript "Molecular architecture underlying fluid absorption by the developing inner ear" to *eLife*. Your article has been favorably evaluated by a Senior Editor, a Reviewing Editor and three expert reviewers. As you will see, all of the reviewers were impressed with the importance and novelty of your work, but they also had multiple comments for improving the study and the manuscript. The three reviews are appended to this letter.

In our on-line discussions among the reviewers, one over-arching critique was that more validation would strengthen the study, and in particular would strengthen (or not) the model in Figure 9. If other investigators have immunolocalized some of the relevant proteins, then a Table summarizing the existing data would be helpful. To give you the flavor of our discussion, an excerpt follows:

"As for validation, there are two major transcriptomics approaches, single cell and microarray. To some degree, they could be seen as cross-checking each other, but the authors don't do that. That seems a minimum requirement; although they can't distinguish RRCs and MRCs in the microarray data, they can get a rough developmental progression in the single cell data. Since both approaches are genome-wide, they should cross-check all of their data to see how well the two datasets compare. And then they should take a representative set of transcripts and do RT-qPCR to confirm trends… Only then should they discuss their model, and they should discuss it as such only if they have localization data to position key molecules on apical or basolateral sides. They need to show that the v-type ATPase is on the same side of the epithelium as SLC26A4 and that the Na^+^/K^+^ ATPase is on the basolateral side. They may also need to show that all of the transporters are localized in the same cell type; they comment that some (*Scnn1b, Kcnq1*, and *Atp1a1*) and say they are not in MRCs, but the others are. The model in Figure 9 only works if all the transporters are in the same cell type, so just detecting them in enolymphatic sac is not enough."

A stylistic point regarding gene names/protein names. Please use only the official name (the gene name) throughout, but when first used in the text note if there is an older and still commonly used name (e.g. Barttin). The legend and the figure labels in Figure 7 are an example of the challenges for the reader. The image has "Barttin" but the legend has "Bsnd".

The reviewers' comments also cover additional suggestions for improving the manuscript. Please use your best judgment in deciding which of these can be reasonably accommodated.

Reviewer #1:

Understanding the functional role of various cell types within the endolymphatic sac of the inner ear is an important endeavor in determining the underlying biology of this structure and the implications in hearing loss. How the endolymphatic sac is able to control fluid volume and how this resorption may affect development is not well understood. The study by Honda and colleagues takes a systematic approach of analyzing the single-cell transcriptional profiles of the dark cells of the endolymphatic sac and those cells with apical microvilli that have abundant mitochondria. The authors identify different groups of cells through principal component analysis and they compare how the expression patterns evolve during development. One major concern with the *Slc26a4* data is the heavy reliance on the Affymetrix chip analysis without validation. The study falls short of showing how or whether these changes affect the function of the endolymphatic sac in the mouse KO. For these reasons, the model of ionic transport and conductances shown in the final figure seems premature. This study could be substantially strengthened by at least validating some of the downregulated transcripts and/or the affected proteins in the mouse KO.

1) In Figure 2, the separation of the two groups is not clear. Why show so many markers for one group and not the other? The omission of multiple genes for the PC2 seems suspect, and the authors could improve this panel by indicating a few of the key genes for the PC2 group. Same for panel C. If there is only one gene known from previous publications, then this information should be clearly stated in the text referring to panel 2B.

2) Subsection “Transcriptomes of single cells from postnatal endolymphatic sac epithelium”, start of last paragraph: "Interestingly" or rather "Unexpectedly"? The reader needs some help in understanding the significance of this finding. Some fleshing out of this paragraph is necessary. The fact that a subset of RRCs is expressing pendrin means either that the model is oversimplified, or this level of pendrin is inconsequential. The log_2_ graph is certainly the conventional way of plotting the data, but what is the difference in fold expression in this case? It would be helpful to state this number in the text.

3) Subsection “Gene array analysis of the endolymphatic sac in *Slc26a4*^Δ/+^ and *Slc26a4*^Δ/Δ^ mice”, last paragraph: 525 genes increased in wild type cells? What is meant here?

4) Although there are some differences in Figure 8 that are indicated as statistically significant, some of the changes look subtle. It is odd to focus on the one gene (*Foxi1*) that doesn't change. Some validation of the key changes in the mouse mutant endolymphatic sac is necessary to support this analysis. Otherwise this study relies too heavily on the Affimetrix chip technology and lacks any functional data from the mouse KO.

Reviewer #2:

Maintenance of endolymph homeostasis is critical for sensory hair cell function in the inner ear, and yet is under-represented in terms of research effort to understand and treat hearing loss and vestibular disease. This manuscript presents a careful and thorough transcriptomic analysis of cells of the developing endolymphatic sac at different developmental stages. The data add new detail to current descriptions and understanding of the endolymphatic sac, but are also consistent with the earlier literature. The authors propose a model-based on their data and on existing literature-for the uptake of NaCl by mitochondria-rich cells in the endolymphatic sac. They make an interesting comparison of the endolymphatic sac to the cortical collecting duct in the kidney, where they propose that three cell types together perform a similar function, but with a much higher fluid turnover. There is a good discussion of the advantages and limitations of the single-cell transcriptomics approach, and for a few genes, expression data have been validated by immunohistochemistry. The figures are all very clear and the manuscript is well written.

*Reviewer #3:*

This manuscript investigates the underlying molecular basis of the EES disorder, which in some cases is due to mutations of SLC26A4. The authors show that fluid absorption by the endolymphatic sac is indeed partially controlled by the presence of SLC26A4, indeed using mice with *Slc26a4* mutations to show that they may have an enlarged endolymphatic sac because they cannot absorb fluid found in their lumens as fast as control mice can. The authors then use single-cell sequencing to infer the presence of two types of cell in the endolymphatic sac, and determine a proposed trajectory for development of these two cell types from common precursors. Finally, they provide evidence for a variety of transmembrane molecules involved in vectorial transport of ions or water, which allows them to propose a molecular model for NaCl absorption (along with fluid) by the sac.

In general, their conclusions follow from their data. I found the order of presentation of the data to be confusing, and I've commented on that below. Importantly, they present a model for NaCl adsorption by the endolymphatic sac. The model is plausible but, other than SLC26A4, is untested for the role of specific molecules. They did use immunocytochemistry to localize several molecules that they detected in transcriptomics experiments, but they did not do this for all of the components of Figure 9.

Moreover, the connection between fluid absorption (Figure 1) and NaCl absorption (Figure 9) was not made experimentally. Presumably fluid absorption by endolymphatic sac corresponds to water flux that occurs in tandem with NaCl flux-but this was not tested, either by ion-substitution experiments or using inhibitors of the components of their model. In the end, I feel that the data doesn't quite come together into the grand model they have proposed. As they state in the last paragraph, "The numerous new details of our model should be further tested experimentally."

Other major comments:

1) The authors clearly have run up against the typical nomenclature problem where the protein name is radically different from the gene name. This leads to inconsistencies throughout the manuscript. For example, for the central gene SLC26A4/*Slc26a4* (properly italicized in the text), the common name is pendrin. But "pendrin" lacks any connection to the gene name. HUGO recommends (but cannot dictate) that for protein names and especially symbols, one should use the gene name with all caps and no italicization. I agree with that. Eventually we will get there for all genes and their proteins. I recommend that the authors use SLC26A4 to refer to both the gene (italics) and protein (not italics), and that they note that its common name is pendrin just once. The same holds for other genes with radically different common protein names, such as SLC4A1 (AE1), CLCNKB (Clc-Kb), and BSND (Barttin). Use the protein symbols that refer to these proteins in the figures as well as the text. An example of the problem is that Figure 7 use different names than Figure 9, so it's very difficult to compare the figures.

2) The order of presentation of the data in this manuscript makes the story quite confusing. For example, as far as I can tell, the single-cell sequencing data are presented in the order the experiments were done (I'm guessing), so that we get P5>P30>E12.5 and E16.5. While there may be some logic to this order, it doesn't help me understand the data. I would either start with the P30 data and work backwards or start with E12.5 and move forwards. It is also strange that the microarray data with *Slc26a4* hets vs. KOs is not shown right after the functional data. Maybe the right order is a) fluid uptake, b) microarray, c) single cell in temporal order.

---

## [Author Response]

In our on-line discussions among the reviewers, one over-arching critique was that more validation would strengthen the study, and in particular would strengthen (or not) the model in Figure 9.

We thank you and the reviewers for pointing this out and below we address your specific recommendations for validation.

If other investigators have immunolocalized some of the relevant proteins, then a Table summarizing the existing data would be helpful.

We added a new table titled “Summary of immunolocalization of proteins in mitochondria-rich cells” as Figure 9—source data 1. As shown in the table, localization of the critical proteins represented in Figure 9 has been demonstrated in other published studies or our current study.

To give you the flavor of our discussion, an excerpt follows:"As for validation, there are two major transcriptomics approaches, single cell and microarray. To some degree, they could be seen as cross-checking each other, but the authors don't do that. That seems a minimum requirement; although they can't distinguish RRCs and MRCs in the microarray data, they can get a rough developmental progression in the single cell data. Since both approaches are genome-wide, they should cross-check all of their data to see how well the two datasets compare. And then they should take a representative set of transcripts and do RT-qPCR to confirm trends. Only then should they discuss their model, and they should discuss it as such only if they have localization data to position key molecules on apical or basolateral sides.

We have cross-checked the microarray and single-cell RNA-seq data against each other. We compared gene array data of E16.5 *Slc26a4*^Δ/+^ endolymphatic sacs with single-cell RNA-seq data of E16.5 wild-type endolymphatic sacs. In addition, we calculated changes of average expression for representative genes between two time points in each of the two methods. A comparison indicates the two methods show positive correlation for moderately or highly expressed genes. Wider dispersion is seen for genes with lower expression levels, which may reflect unreliability of low-end TPM values and high stochastic variation in single-cell RNA-seq data. For representative genes, the direction and magnitude of the expression changes are consistent between the two data sets We added this data in a new figure: Figure 8—figure supplement 2.

We also assessed gene expression of whole endolymphatic sacs obtained from wild-type mice at ages E12.5 to P90 by RT-qPCR. We compared the temporal expression trends with those within the gene array data of *Slc26a4*^Δ/+^ endolymphatic sacs, and confirmed the trends were similar to each other. We show the details in Figure 8—figure supplement 3.

They need to show that the v-type ATPase is on the same side of the epithelium as SLC26A4 and that the Na^+^/K^+^ ATPase is on the basolateral side.

Immunolocalization of *Slc26a4* and the v-type ATPase subunits *Atp6v0a4* and *Atp6v1b1* have already been published in previous studies. We have summarized the existing data and our data in Figure 9—source data 1. We also added the immunolocalization of *Atp1a1* to Figure 1.

They may also need to show that all of the transporters are localized in the same cell type; they comment that some (Scnn1b, Kcnq1, and Atp1a1) and say they are not in MRCs, but the others are. The model in Figure 9 only works if all the transporters are in the same cell type, so just detecting them in enolymphatic sac is not enough."

Our results of single cell RNA-seq indicate that *Scnn1b, Kcnq1*, and *Atp1a1* are expressed not only in MRCs but in both cell types (see Author response image 1). Since we haven’t validated the localization of *Scnn1b* or *Scnn1a* or *Kcnq1*, we have added question marks to indicate the uncertain localization of the apical Na^+^ channel and the basal K^+^ channel in Figure 9.

A stylistic point regarding gene names/protein names. Please use only the official name (the gene name) throughout, but when first used in the text note if there is an older and still commonly used name (e.g. Barttin). The legend and the figure labels in Figure 7 are an example of the challenges for the reader. The image has "Barttin" but the legend has "Bsnd".

We have changed gene/protein symbols to follow HUGO guidelines.

The reviewers' comments also cover additional suggestions for improving the manuscript. Please use your best judgment in deciding which of these can be reasonably accommodated.Reviewer #1:Understanding the functional role of various cell types within the endolymphatic sac of the inner ear is an important endeavor in determining the underlying biology of this structure and the implications in hearing loss. How the endolymphatic sac is able to control fluid volume and how this resorption may affect development is not well understood. The study by Honda and colleagues takes a systematic approach of analyzing the single-cell transcriptional profiles of the dark cells of the endolymphatic sac and those cells with apical microvilli that have abundant mitochondria. The authors identify different groups of cells through principal component analysis and they compare how the expression patterns evolve during development. One major concern with the Slc26a4 data is the heavy reliance on the Affymetrix chip analysis without validation. The study falls short of showing how or whether these changes affect the function of the endolymphatic sac in the mouse KO. For these reasons, the model of ionic transport and conductances shown in the final figure seems premature. This study could be substantially strengthened by at least validating some of the downregulated transcripts and/or the affected proteins in the mouse KO.1) In Figure 2, the separation of the two groups is not clear. Why show so many markers for one group and not the other? The omission of multiple genes for the PC2 seems suspect, and the authors could improve this panel by indicating a few of the key genes for the PC2 group. Same for panel C. If there is only one gene known from previous publications, then this information should be clearly stated in the text referring to panel 2B.

We assume the reviewer was referring to RRCs as ‘PC2’ here. According to previous publications, *Notch1* is the only known marker of embryonic non-MRCs, as described in the first paragraph of the subsection “Transcriptomes of single cells from postnatal endolymphatic sac epithelium”. There is no literature to support *Notch1* as a marker of the postnatal RRC population, which was why we did not mention it is an “RRC marker”. We have added a description to clarify this.

2) Subsection “Transcriptomes of single cells from postnatal endolymphatic sac epithelium”, start of last paragraph: "Interestingly" or rather "Unexpectedly"? The reader needs some help in understanding the significance of this finding. Some fleshing out of this paragraph is necessary. The fact that a subset of RRCs is expressing pendrin means either that the model is oversimplified, or this level of pendrin is inconsequential. The log_2_ graph is certainly the conventional way of plotting the data, but what is the difference in fold expression in this case? It would be helpful to state this number in the text.

Further experiments are necessary to interpret the significance of pendrin expression in a subset of RRCs, but those experiments are beyond the scope of this study.

We thank the reviewer for advising us to state the difference in fold-change in expression. We calculated mean expression levels of each cell group (MRCs, *Slc26a4*-positive RRCs, *Slc26a4*-negative RRCs) at both ages, and then calculated the difference in fold-change in expression between MRCs and *Slc26a4*-positive RRCs. Author response image 2 shows the distribution of expression levels in each cell group (MRCs, *Slc26a4*-positive RRCs, or *Slc26a4*-negative RRCs), where each dotted line indicates the mean expression level for each group.

**Author response image 2. respfig2:** 

The values of the mean levels are shown in the following table.

AgeP5P30Cell groupMRCRRC (*Slc26a4*+)RRC (*Slc26a4*-)MRCRRC (*Slc26a4*+)RRC (*Slc26a4*-)Mean, log_2_ (nTPM)11.187.380.8111.517.330.84

Finally, we added the following sentence:

“The mean expression levels of *Slc26a4* for the *Slc26a4*-positive RRCs (> 5 log_2_ (nTPM)) are 14- to 18-fold lower than those for MRCs.”

3) Subsection “Gene array analysis of the endolymphatic sac in Slc26a4^Δ/+^ and Slc26a4^Δ/Δ^ mice”, last paragraph: 525 genes increased in wild type cells? What is meant here?

Those genes increased regardless of genotype. We have changed the description to:

“Signal intensities for 680 probes (525 genes) significantly increased during the time interval in *Slc26a4*^Δ/+^, *Slc26a4*^Δ/Δ^, or both.”

4) Although there are some differences in Figure 8 that are indicated as statistically significant, some of the changes look subtle. It is odd to focus on the one gene (Foxi1) that doesn't change. Some validation of the key changes in the mouse mutant endolymphatic sac is necessary to support this analysis. Otherwise this study relies too heavily on the Affimetrix chip technology and lacks any functional data from the mouse KO.

We focused on *Foxi1* because we wanted to confirm there was no difference of MRC/RRC ratio between *Slc26a4*^Δ/+^ and *Slc26a4*^Δ/Δ^. Since the ratio doesn’t differ between the two genotypes, gene expression changes at a tissue level represent cellular-level expression changes, not a change in proportion in cell populations, as we stated in the third paragraph of the subsection “Gene array analysis of the endolymphatic sac in *Slc26a4*^Δ/+^ and *Slc26a4*^Δ/Δ^ mice”.

We acknowledge that validation of expression levels of key proteins between *Slc26a4*^Δ/+^ and *Slc26a4*^Δ/Δ^ would illuminate the mechanism of pathogenesis. However, whether or not some proteins (i.e. Clcnkb or Bsnd) are downregulated in *Slc26a4*^Δ/Δ^ doesn’t change our conclusion.

Reviewer #3:In general, their conclusions follow from their data. I found the order of presentation of the data to be confusing, and I've commented on that below. Importantly, they present a model for NaCl adsorption by the endolymphatic sac. The model is plausible but, other than SLC26A4, is untested for the role of specific molecules. They did use immunocytochemistry to localize several molecules that they detected in transcriptomics experiments, but they did not do this for all of the components of Figure 9.Moreover, the connection between fluid absorption (Figure 1) and NaCl absorption (Figure 9) was not made experimentally. Presumably fluid absorption by endolymphatic sac corresponds to water flux that occurs in tandem with NaCl flux-but this was not tested, either by ion-substitution experiments or using inhibitors of the components of their model. In the end, I feel that the data doesn't quite come together into the grand model they have proposed. As they state in the last paragraph, "The numerous new details of our model should be further tested experimentally."

We express thanks for the reviewer’s comments. Immunolocalization of *Slc26a4, Atp6v0a4*, and *Atp6v1b1* have already been published in previous studies. We have summarized the existing data including that from this study in Figure 9—source data 1. We also added the immunohistochemistry of *Atp1a1* to Figure 1. Unfortunately, we failed to confirm localization of *Scnn1a* since we could not find a good antibody.

In addition, we performed in vivofunctional studies using ouabain, benzamil, bafilomycin, S3226 and gadoliniium. Those drugs inhibit function of Na^+^/K^+^ ATPase, ENaC, v-type ATPase, Na^+^/H^+^ exchanger NHE3, and non-selective cation channels, respectively. The result indicates that fluid absorption of E14.5 endolymphatic sac was sensitive to ouabain and gadolinium but insensitive to benzamil, bafilomycin and S3226. Furthermore, we have added immunohistochemistry demonstrating the expression of *Atp1a1* in the basolateral membranes of endolymphatic sac epithelial cells. All of these data are now included in Figure 1.

Other major comments:1) The authors clearly have run up against the typical nomenclature problem where the protein name is radically different from the gene name. This leads to inconsistencies throughout the manuscript. For example, for the central gene SLC26A4/Slc26a4 (properly italicized in the text), the common name is pendrin. But "pendrin" lacks any connection to the gene name. HUGO recommends (but cannot dictate) that for protein names and especially symbols, one should use the gene name with all caps and no italicization. I agree with that. Eventually we will get there for all genes and their proteins. I recommend that the authors use SLC26A4 to refer to both the gene (italics) and protein (not italics), and that they note that its common name is pendrin just once. The same holds for other genes with radically different common protein names, such as SLC4A1 (AE1), CLCNKB (Clc-Kb), and BSND (Barttin). Use the protein symbols that refer to these proteins in the figures as well as the text. An example of the problem is that Figure 7 use different names than Figure 9, so it's very difficult to compare the figures.

We corrected all gene and protein names according to the HUGO gene nomenclature guideline.

2) The order of presentation of the data in this manuscript makes the story quite confusing. For example, as far as I can tell, the single-cell sequencing data are presented in the order the experiments were done (I'm guessing), so that we get P5>P30>E12.5 and E16.5. While there may be some logic to this order, it doesn't help me understand the data. I would either start with the P30 data and work backwards or start with E12.5 and move forwards. It is also strange that the microarray data with Slc26a4 hets vs. KOs is not shown right after the functional data. Maybe the right order is a) fluid uptake, b) microarray, c) single cell in temporal order.

We appreciate the reviewer’s suggestion. We also struggled with the order of presentation and respectfully prefer to retain the current organization.